# TeMo: Temperature Modulation for Multimodal Contrastive Learning

## Abstract

Contrastive learning approaches achieve strong performance by training models to bring similar samples closer while pushing dissimilar samples apart. A crucial component of contrastive learning is the temperature hyperparameter $\tau$, which controls the penalty strength applied to negative samples. However, most existing methods either fix this hyperparameter or learn a global value during training. In this paper, we introduce TeMo, Temperature Modulation framework, a similarity-based modulation approach that adaptively adjusts the temperature for each positive-negative pair according to their similarity, enabling more fine-grained multimodal contrastive learning. Our approach seamlessly integrates temperature-modulated multimodal and unimodal losses with the standard multimodal contrastive loss by gradually transitioning between them. This design allows the model to capture both coarse- and fine-grained semantics at different training stages. Extensive experiments demonstrate that each component of TeMo consistently enhances performance across diverse zero-shot retrieval and classification tasks, establishing new state-of-the-art results.

## 1 Introduction

Contrastive Learning (CL) has become one of the most relevant self-supervised representation learning paradigms, allowing the training of high performing models on large amounts of unlabeled data. It facilitates the learning of robust unimodal He et al. (2019); Chen et al. (2020a) and multimodal Radford et al. (2021); Cherti et al. (2023); Zhai et al. (2023); Xu et al. (2024) representations by encouraging matching pairs to be close in the embedding space, while pushing nonmatching pairs farther apart. One of the most important parameters that affect the structure of the learned representations is the temperature $\tau$. In unimodal learning, $\tau$ plays a key role in controlling the spread of concepts Wang & Liu (2021); Kukleva et al. (2023): higher temperature tends to produce a tighter cluster of samples, learning fine-grained semantic information of each sample, while a lower temperature leads to a more uniform representation space by learning coarser class-wise semantics. In multimodal learning, however, $\tau$ mainly controls the so-called modality gap Liang et al. (2022); Wang & Liu (2021) that separates the vision and language modalities. Varying the temperature directly affects the gap: a high temperature closes it and can even eliminate it, whereas a lower temperature drives the modalities farther apart Liang et al. (2022). Whereas in this work, we integrate both multimodal and unimodal losses within a unified temperature modulation framework, enabling fine-grained control over the learning dynamics of both multimodal and unimodal representations. We show that this joint modulation not only enhances alignment between modalities by reducing the modality gap, but also improves the structure of unimodal embeddings.

Most of the existing methods Mu et al. (2022); Radford et al. (2021); Tang et al. (2025); He et al. (2020); Chen et al. (2020b) use a global learnable temperature parameter shared across all pairs. Consequently, the model seeks an overall balance among different semantic classes by applying similar repulsive forces to diverse negative samples. However, in certain scenarios, such as long-tail datasets, it is more desirable to allow fine-grained control over local structures and underrepresented classes in the embedding space Kukleva et al. (2023). Building on this, variable temperature schemes have been proposed to improve representation learning based on temperature alternations Wang & Liu (2021); Qiu et al. (2024); Kim & Kim (2025); Li et al. (2023); Wang et al. (2020); Zhang et al. (2021); Khaertdinov et al. (2022); Kukleva et al. (2023).

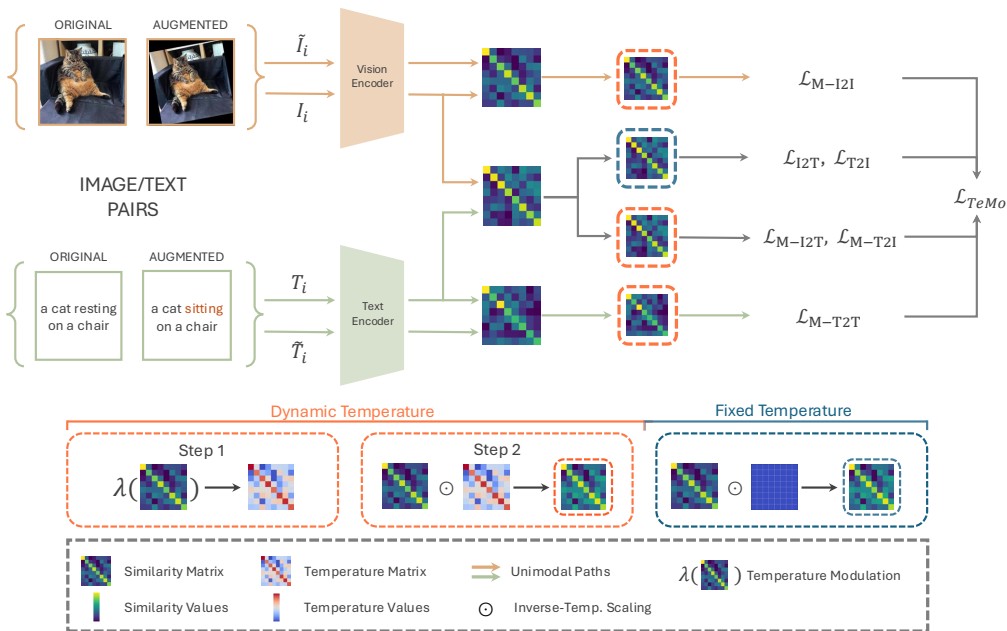

Figure 1: **High-level overview of TeMo.** A batch of image–text pairs and their augmentations are fed through a vision and a text encoder to generate embeddings $\mathbf{I}_i, \tilde{\mathbf{I}}_i, \mathbf{T}_i, \tilde{\mathbf{T}}_i$. A fixed-temperature branch computes the standard crossmodal $\mathcal{L}_{\mathrm{MM}}$. In parallel, similarity matrices for image-to-text, text-to-image, image-to-image, and text-to-text are passed through the temperature modulation mechanism, mapping similarities in [0,1] to temperatures in $[\tau_{\min}, \tau_{\min} + \tau_\alpha]$. The modulated temperatures are used to compute the crossmodal temperature modulated loss $\mathcal{L}_{\mathrm{M-MM}}$ and the two unimodal losses $\mathcal{L}_{\mathrm{M-I2I}}$ and $\mathcal{L}_{\mathrm{M-T2T}}$. Lastly, a quadratic scheduler blends the standard loss with the set of temperature-modulated losses, producing the final objective $\mathcal{L}_{\mathrm{TeMo}}$.

To this end, we propose Temperature Modulation (TeMo), a novel multimodal contrastive learning (CL) framework that leverages combined multimodal and unimodal temperature modulation to enhance multimodal representation learning. Unlike prior methods that use a globally assigned temperature Kukleva et al. (2023); Qiu et al. (2024); Wang & Liu (2021), TeMo adaptively modulates the temperature for *each* positive-negative pair individually. The per-pair temperature strategy enables more precise control over the contrastive objective, allowing the model to dynamically adjust its learning signals based on the similarity of individual paired samples. Additionally, temperature-modulated unimodal losses improve local structure within each modality. We integrate the modulated multimodal and unimodal contrastive losses into a standard CL framework through progressive scheduling. This scheduling enables the model to initially capture instance-level semantic details using a lower temperature, while our adaptive temperature modulation gradually guides sample representations toward coarser semantic groupings.

We evaluate TeMo on the two standard tasks: multimodal zero-shot retrieval on MSCOCO and Flickr30k, and zero-shot image classification on CIFAR10, CIFAR100, and ImageNet-1k, while pretraining the model on the large-scale CC3M or CC12M dataset. TeMo consistently outperforms existing temperature-adaptation baselines for both tasks and across datasets. We summarize our contribution as follows:

- We propose TeMo, a novel temperature modulation framework for multimodal contrastive learning, which introduces a *per pair* temperature modulation based on the similarity of each pair for both unimodal and multimodal losses, enabling more precise control over the learned representation space;

- We show that combining multimodal and unimodal contrastive losses is particularly effective when used with our temperature modulation approach;

- We provide an in-depth evaluation of the characteristics of the proposed system and show that TeMo outperforms prior temperature adaptation approaches on zero-shot retrieval and classification benchmarks.

## 2 RELATED WORK

**Unimodal CL** learns robust representations from single modalities (e.g., images or text) by aligning augmented views of the same input using the InfoNCE loss Oord et al. (2018). Prominent methods include MoCo He et al. (2020); Chen et al. (2020b), which maintains a momentum-based queue for negatives, and SimCLR Chen et al. (2020a), which utilizes samples from the same training batch. The performance of unimodal CL critically depends on the number of negative samples Kalantidis et al. (2020); Zhang et al. (2022); Yeh et al. (2021). In our approach, we adopt the batch-wise negative sampling strategy of SimCLR.

**Multimodal CL** extends contrastive learning to multiple modalities, such as images and texts, aiming to align corresponding pairs and distinguish them from unrelated ones. Methods like CLIP Radford et al. (2021) optimize a cross-modal InfoNCE objective and demonstrate remarkable generalization in downstream tasks such as zero-shot retrieval and classification. Recent work proposes a wide range of enhancements: DeCLIP Li et al. (2021) and SLIP Mu et al. (2022) incorporate vision-specific unimodal self-supervision; CWCL Srinivasa et al. (2023) proposes a new loss function that uses continuous, rather than binary, similarity scores; FILIP Yao et al. (2021) and DeFILIP Cui et al. (2022) introduce fine-grained late interaction between the two modalities; CyCLIP Goel et al. (2022) enforces geometric consistency through the usage of two additional objectives on top of the standard InfoNCE loss; SigLIP Zhai et al. (2023) and SigLIP2 Tschannen et al. (2025) explore alternative contrastive objectives; SoftCLIP Gao et al. (2024) relaxes the strict one-to-one alignment assumption by introducing soft cross-modal targets derived from intra-modal similarities; SiLC Naeem et al. (2024) enhances representation quality via self-distillation; LaCLIP Fan et al. (2023) leverages language-centric augmentation and TULIP Tang et al. (2025) enhances fine-grained visual understanding while preserving semantic alignment by combining generative data augmentation, intra-modal contrastive learning, and reconstruction-based regularization. In our work, we integrate multimodal and unimodal contrastive losses through a novel temperature modulation framework.

**Temperature in CL.** Recent works Wang & Liu (2021); Qiu et al. (2024); Kim & Kim (2025); Li et al. (2023); Wang et al. (2020); Zhang et al. (2021); Khaertdinov et al. (2022); Kukleva et al. (2023) have shown that the temperature parameter $\tau$ is crucial for the formation of the embedding space, both in the unimodal and multimodal setting. Wang & Liu (2021) demonstrated that the unimodal CL objective with high temperature focuses more on the hard negatives, samples which are very similar to the anchor but semantically different. In contrast, when a low temperature is used, the loss treats all negatives more equally, encouraging the formation of larger clusters Kukleva et al. (2023).

This implicit focusing mechanism relates closely to Hard Negative Mining (HNM). While temperature modulates the impact of negatives within the loss calculation, HNM explicitly intervenes at the data level to alter the sampling distribution. Early HNM approaches focused on sampling strategies to bias training toward difficult examples Robinson et al. (2020), effectively decoupling the mining process from the loss function. Wang & Liu (2021) proved that low-temperature InfoNCE acts asymptotically like a triplet loss with implicit HNM. However, relying solely on a scalar $\tau$ couples the alignment of positive pairs with the repulsion of negative pairs. To resolve this, Yeh et al. (2021) proposed decoupled learning by removing the positive term from the denominator, allowing for independent control over positive alignment and negative mining. Consequently, a complementary relationship emerges: while HNM ensures the model is *exposed* to difficult samples, adaptive temperature mechanisms are required to strictly *calibrate* their gradient contribution, preventing instability from outliers while maintaining focus on the decision boundary.

To this end, various adaptive frameworks have been proposed beyond a fixed global $\tau$. In the unimodal setting, DySTreSS Manna et al. (2025) sets a pairwise temperature as a cosine function of similarity, pushing highly similar pairs less and dissimilar pairs more. Temperature Schedules Kukleva et al. (2023) replace constant $\tau$ with a cosine schedule, periodically switching between instance and group-wise discrimination for long-tailed data. MACL Huang et al. (2023) makes $\tau$ alignment-aware, while Dynamic Temperature Scaling Khaertdinov et al. (2022) uses a frozen aux-

iliary encoder to map instance-level similarities into temperatures for negative pairs. Furthermore, Temperature-Free CL Kim & Kim (2025) removes the hyperparameter entirely by replacing $s/\tau$ with a monotone log-odds mapping. In parallel, Yaras et al. (2024) provides valuable theoretical insights into how temperature modulation specifically affects the modality gap. While our work aligns with their findings, we distinguish our approach by explicitly tackling this gap through additional constraints rather than temperature alone. Building on these insights, we extend similarity-driven temperature modulation to the multimodal setting, incorporating self-supervision to manage the modality gap without additional training overhead.

## 3 TeMo Method

In this section, we introduce our proposed framework, TeMo. We first provide a high-level overview, followed by a discussion of various contrastive losses in the preliminaries. We then describe each component of the framework in detail in the subsequent paragraphs.

**Overview.** The TeMo framework comprises three main components: (1) a standard multimodal InfoNCE loss, (2) a modulated multimodal and unimodal InfoNCE losses, and (3) a progressive scheduler that gradually adjusts the balance between standard and temperature-modulated losses during training. An overview of the framework is shown in Figure 1. Initially, the standard InfoNCE loss enables the model to capture instance-specific semantic information using a low-temperature parameter, establishing global uniformity. For temperature modulation, we leverage the theoretical trade-off between alignment and uniformity Wang & Liu (2021): we assign lower temperatures to dissimilar pairs to enforce sharper separation, while increasing temperatures for high-similarity pairs to prioritize semantic alignment. Finally, using our progressive scheduler, we gradually increase the influence of the modulated losses. This acts as a coarse-to-fine optimization strategy, shifting focus from early-stage uniformity toward refined local alignment, enabling the model to precisely control multimodal and unimodal relationships within the embedding space.

**Notation.** Let $\mathcal{D} = (\mathbf{I}_i, \mathbf{T}_i)_{i=1}^N$ denote a multimodal dataset consisting of $N$ paired samples. We denote with $\mathbf{I} = \{\mathbf{I}_1, \ldots, \mathbf{I}_N\}$ a random batch of images from $\mathcal{D}$, by $\mathbf{I}_i$ an individual image within the batch, by $\mathbf{T} = \{\mathbf{T}_1, \ldots, \mathbf{T}_N\}$ the corresponding batch of paired text samples, and by $\mathbf{T}_i$ an individual text within the batch of texts $\mathbf{T}$. The augmented versions of the image and text batches are denoted by $\tilde{\mathbf{I}} = \{\tilde{\mathbf{I}}_1, \ldots, \tilde{\mathbf{I}}_N\}$ and $\tilde{\mathbf{T}} = \{\tilde{\mathbf{T}}_1, \ldots, \tilde{\mathbf{T}}_N\}$ respectively, where each $\tilde{\mathbf{I}}_i \sim \mathcal{A}(\mathbf{I}_i)$ is a random augmentation applied to $\mathbf{I}_i$ and each $\tilde{\mathbf{T}}_i$ is a rephrased version of $\mathbf{T}_i$ using a large language model. We denote by $\mathcal{T} = \{\tau_{ij}\}_{i,j=1}^N$ the set of temperature values used to modulate the similarity scores between pairs. For simplicity, we will refer to the batches of the embeddings as $\mathbf{I}, \tilde{\mathbf{I}}$ and $\mathbf{T}, \tilde{\mathbf{T}}$. Lastly, we define the similarity between two embeddings $\mathbf{x}$ and $\mathbf{y}$ as $\langle \bar{\mathbf{x}}, \bar{\mathbf{y}} \rangle$, where $\bar{\cdot}$ denotes an L2 normalized vector.

### 3.1 Preliminaries

**Contrastive Loss.** In this work, we adopt the InfoNCE loss Khosla et al. (2020); Oord et al. (2018) as our contrastive objective, that learns to place similar pairs in a close proximity, while dissimilar pairs farther apart. Given a sample $\mathbf{x_a}$, called an anchor, a set of samples $X$ containing negative samples and one corresponding positive sample $\mathbf{x_p}$, and set of temperature parameters $\mathcal{T}$, we define the InfoNCE loss as follows:

$$\mathcal{I}(\mathbf{x_a}, X; \mathcal{T}) = -\log \frac{\exp(\langle \bar{\mathbf{x}}_\mathbf{a}, \bar{\mathbf{x}}_\mathbf{p} \rangle / \tau_{\mathbf{ap}})}{\sum_{\mathbf{x_k} \in X} \exp(\langle \bar{\mathbf{x}}_\mathbf{a}, \bar{\mathbf{x}}_\mathbf{k} \rangle / \tau_{\mathbf{ak}})}. \tag{1}$$

In methods such as MoCo or SimCLR, the set of the temperature parameters $\mathcal{T}$ is represented by only one unique value, the constant or learnable temperature for all the pairs in $\mathcal{D}$.

**Unimodal Contrastive Loss.** In the classical unimodal CL scenario, where labels for individual instances are unavailable, the positive set comprises augmentations of the anchor. Given a batch of images $\mathbf{I}$, its corresponding augmentations $\tilde{\mathbf{I}}$, and the temperature set $\mathcal{T}$, for any image $\mathbf{I}_i \in \mathbf{I}$, the unimodal InfoNCE loss can be defined as follows:

$$\mathcal{L}_{\text{I2I}}(\mathbf{I}, \tilde{\mathbf{I}}; \mathcal{T}) = \frac{1}{N} \sum_{i=1}^N \mathcal{I}(\mathbf{I}_i, \tilde{\mathbf{I}}; \mathcal{T}). \tag{2}$$

Similarly, the unimodal formulation can easily be extended to other modalities as well, such as text, where the objective is applied on texts and their augmentations.

**Multimodal Contrastive Loss.** The unimodal InfoNCE objective can be extended to the multimodal setting. Unlike the unimodal case, where both the anchor and the positive sample belong to the same modality, the multimodal objective encourages alignment between representations from different modalities. In this context, the anchor and its corresponding positive come from paired data across modalities. Given a multimodal dataset $\mathcal{D}$ consisting of paired samples, the multimodal InfoNCE loss can be defined as follows:

$$\mathcal{L}_{\text{I2T}}(\mathbf{I}, \mathbf{T}, \mathcal{T}) = \frac{1}{N} \sum_{i=1}^{N} \mathcal{I}(\mathbf{I}_i, \mathbf{T}, \mathcal{T}),$$

$$\mathcal{L}_{\text{MM}}(\mathbf{I}, \mathbf{T}; \mathcal{T}) = \frac{1}{2} \sum_{i=1}^{N} \left[ \mathcal{L}_{\text{I2T}}(\mathbf{I}_i, \mathbf{T}; \mathcal{T}) + \mathcal{L}_{\text{T2I}}(\mathbf{T}_i, \mathbf{I}; \mathcal{T}) \right],$$

(3)

where $\mathcal{L}_{\text{I2T}}$ denotes the image-to-text loss, aligning each image with its corresponding text. For brevity, we omit the explicit form of $\mathcal{L}_{\text{T2I}}$, which is defined in a similar manner, aligning each text with its paired image.

## 3.2 TEMPERATURE MODULATION FRAMEWORK

**Temperature Modulation.** At the core of the framework is the use of a *per pair* temperature $\tau_{ij}$. TeMo computes the temperature by using the similarity between paired samples as a proxy. We define temperature modulation as follows:

$$\lambda(\bar{\mathbf{x}}, \bar{\mathbf{y}}) = \tau_{\min} + \tau_{\alpha} \sqrt{\text{sim}(\bar{\mathbf{x}}, \bar{\mathbf{y}})},$$

(4)

where $\tau_{\min}$ denotes the minimum temperature, and $\tau_{\alpha}$ is a hyperparameter that controls the maximum temperature during training. $\lambda(\bar{\mathbf{x}}, \bar{\mathbf{y}})$ maps similarity scores, originally in the range $[0.0, 1.0]$, to a temperature value in the interval $[\tau_{\min}, \tau_{\min} + \tau_{\alpha}]$. To ensure a smooth transition between similarity values and temperatures, we apply a square root to the similarity score. Since the square root function is monotonically increasing, highly similar pairs receive higher temperatures, reducing the penalty, while less similar pairs are penalized more aggressively.

The modulation is performed in an element-wise manner across the similarity matrix. As a result, the computed temperature values can be viewed as a matrix of adaptive temperatures specific to each input pair.

**Unimodal Temperature Modulation.** Given a batch $\mathcal{X}$, either images or texts, of unimodal samples and their augmentations $\tilde{\mathcal{X}}$, we define the modulated unimodal contrastive loss as follows:

$$\mathcal{L}_{\text{M-}\mathcal{X}2\tilde{\mathcal{X}}}(\mathcal{X}, \tilde{\mathcal{X}}; \mathcal{T}_{\mathcal{X}2\tilde{\mathcal{X}}}) = \frac{1}{N} \sum_{i=1}^{N} \mathcal{I}\left(\mathcal{X}_i, \tilde{\mathcal{X}}; \{\tau_{ij}\}_{j=1}^{N}\right),$$

(5)

where each anchor $\mathcal{X}_i$ and sample $\tilde{\mathcal{X}}_j$ in the batch is associated with a *per pair* temperature $\tau_{ij}$, as defined in Equation 4.

**Multimodal Temperature Modulation.** Given a batch $\mathbf{I}$ of images and their corresponding texts $\mathbf{T}$, we define the two modulated crossmodal losses as follows:

$$\mathcal{L}_{\text{M-I2T}}(\mathbf{I}, \mathbf{T}; \mathcal{T}_{\text{I2T}}) = \frac{1}{N} \sum_{i=1}^{N} \mathcal{I}\left(\mathbf{I}_i, \mathbf{T}; \{\tau_{ij}\}_{j=1}^{N}\right),$$

$$\mathcal{L}_{\text{M-T2I}}(\mathbf{T}, \mathbf{I}; \mathcal{T}_{\text{T2I}}) = \frac{1}{N} \sum_{i=1}^{N} \mathcal{I}\left(\mathbf{T}_i, \mathbf{I}; \{\tau_{ij}\}_{j=1}^{N}\right),$$

(6)

where $\mathcal{T}_{\text{I2T}}$ is the temperature matrix derived from the image-to-text similarities, and $\mathcal{T}_{\text{T2I}}$ its transpose, corresponds to the temperature matrix derived from text-to-image similarities. The multimodal temperature-modulated loss $\mathcal{L}_{\text{M-MM}}$ is therefore defined as follows:

$$\mathcal{L}_{\text{M-MM}}(\mathbf{I}, \mathbf{T}; \{\mathcal{T}_{\text{I2T}}, \mathcal{T}_{\text{T2I}}\}) = \frac{1}{2}(\mathcal{L}_{\text{M-I2T}} + \mathcal{L}_{\text{M-T2I}}).$$

(7)

For clarity and space, we omit the explicit inputs to $\mathcal{L}_{\text{M-I2T}}$ and $\mathcal{L}_{\text{M-T2I}}$ in the notation above, though they follow the same parameterization as in Equation 6.

**TeMo.** We propose to combine the standard InfoNCE loss with the temperature-modulated multimodal and unimodal InfoNCE losses. The overall objective is defined as follows:

$$\mathcal{L}_{\text{TeMo}} = \alpha\mathcal{L}_{\text{MM}} + \beta(\mathcal{L}_{\text{M-MM}} + \mathcal{L}_{\text{M-I2I}} + \mathcal{L}_{\text{M-T2T}}), \tag{8}$$

where $\mathcal{L}_{\text{MM}}$ refers to the standard multimodal InfoNCE loss, $\mathcal{L}_{\text{M-MM}}$ to the modulated multimodal loss, and $\mathcal{L}_{\text{M-I2I}}$ and $\mathcal{L}_{\text{M-T2T}}$ to the modulated unimodal losses. Additionally, TeMo employs a progressive scheduler to control how the losses are integrated during training. Specifically, we adopt a quadratic blend with $\alpha = (1-t)^2$, $\beta = t^2$, where $t$ represents the normalized training step, $t \in [0, 1]$. At the beginning of training, the standard InfoNCE loss dominates, providing the initial contrastive signal. As training progresses, the emphasis gradually shifts toward more precise alignment and reinforcement of the learned structure. In particular, unimodal image and text losses regulate their respective representation spaces by enforcing better structure through individual pairwise similarities. Meanwhile, introducing fine-grained control over the multimodal temperature helps reduce the modality gap. To this end, combining multimodal and unimodal contrastive losses allows for more precise control over the learned semantics in both multimodal and unimodal embeddings. For a high-level overview of the algorithm, see Appendix B.

## 4 EXPERIMENTS

This section validates our proposed TeMo framework. First, we compare our method to the state-of-the-art work on both retrieval and classification tasks in Section 4.1. Then, we analyze each component separately to show the importance and connections of each to the improved performance. Finally, we analyze learned representations with respect to visual modality and individual temperatures.

**Datasets.** Conceptual Captions 3M (CC3M) Sharma et al. (2018) is a large-scale dataset of image-text pairs collected from the internet. The training split contains approximately 2.9M image-text pairs. Conceptual Captions 12M (CC12M) Changpinyo et al. (2021) is an extended version of CC3M, comprising around 12 million image-text pairs. MSCOCO is a large-scale dataset commonly used for tasks like object detection and image captioning. Following previous studies Yuan et al. (2022); Goel et al. (2022), we leverage the Karpathy split Karpathy & Fei-Fei (2015) for evaluation, which includes 5,000 image-text pairs each for the testing split. Flickr30k Young et al. (2014) is a small multimodal dataset which includes approximately 158,000 captions paired with 30,000 images. Similarly, we leverage the Karpathy split for evaluation. CIFAR10 and CIFAR100 Krizhevsky et al. (2009) comprise 60,000 low-resolution images each, organized into 10 and 100 object classes, respectively. ImageNet-1k Deng et al. (2009) contains around 1.2 M training and 50k validation high-resolution images organized into 1k object classes.

**Evaluation.** We evaluate TeMo on the tasks of multimodal zero-shot retrieval and zero-shot image classification. For all experiments on CC3M, we first initialize the vision and text encoders *separately* rather than using a joint CLIP backbone, following Qiu et al. (2024), and then fine-tune them on CC3M. For CC12M, we build upon the official SLIP Mu et al. (2021) repository. Retrieval performance on MSCOCO and Flickr30k is measured using Recall at ranks 1, 5, and 10 (R@1, R@5, R@10) for both text-to-image and image-to-text, full evaluations are in the supplementary material. For zero-shot classification on CIFAR10, CIFAR100 and ImageNet-1k, we report Top-K classification accuracy at ranks 1, 3, and 5 (T@1, T@3, T@5). To perform zero-shot classification, we follow the standard CLIP protocol by using class names as textual prompts and selecting the class whose text embedding has the highest similarity to the image embedding.

**Implementation Details:** For CC3M experiments, we leverage a ResNet50 He et al. (2016) or ViT-B/16 Dosovitskiy et al. (2020) pre-trained on ImageNet-1k as the vision encoder and a pre-trained DistilBERT Sanh et al. (2019) model as the text encoder, while for CC12M we train from scratch. We pre-generate five paraphrases for every caption in the datasets using a PEGASUS Zhang et al. (2019) model fine-tuned for paraphrasing and sample one of the variants uniformly at each training step. For the InfoNCE baseline, we use a fixed temperature parameter and set its value to $\tau = 0.01$ according to Qiu et al. (2024). For the TS baseline, we vary the temperature in $\tau \in [0.01, 0.05]$ across five periods. For SLIP, we adopt a learnable temperature initialized at 0.07. Furthermore, we

extend DySTreSS Manna et al. (2025) to the multimodal setting and re-implement CWCL Srinivasa et al. (2023), as no public code is available. To ensure a fair comparison, we train the models using the same optimization settings and backbone architectures as TeMo. To ensure a fair comparison with SoftCLIP, we trained both the InfoNCE baseline and TeMo from scratch on CC3M. In our method, we modulate the temperature according to Equation 4 and set $\tau_{min} = 0.01$ and $\tau_\alpha = 0.04$, which effectively gives us a range of temperatures between $[0.01, 0.05]$. We follow standard practice in fixing $\tau_{min}$ (commonly set to 0.01 or 0.07) and do not tune it, so only $\tau_\alpha$ is treated as a tunable hyperparameter. Lastly, for training, we use a batch size of 2048 on CC3M and 4096 on CC12M.

## 4.1 COMPARISON TO SOTA

In Table 1, we compare TeMo to the standard InfoNCE objective, Temperature Schedules Kukleva et al. (2023), DySTreSS Manna et al. (2025) and CWCL Srinivasa et al. (2023) across both multimodal retrieval and zero-shot classification tasks. Using a lightweight ResNet-50, TeMo improves MSCOCO IR@1/TR@1 by +1.64/+1.76% and Flickr30k IR@1/TR@1 by +2.04/+2.4% relative to InfoNCE. Moreover, we observe significant improvement over TS, DySTreSS and CWCL frameworks. Note that the original TS framework is evaluated only on ResNet architectures, and we observe a degradation in retrieval performance when using a ViT backbone. In contrast, our TeMo framework consistently improves performance with the larger ViT-B/16 backbone, highlighting its robustness across different architectures.

Furthermore, evaluating TeMo on zero-shot classification reveals significant improvements of over 10% on both CIFAR10 and CIFAR100. Specifically, with a ResNet-50 backbone, TeMo achieves notable gains, improving Top-1 accuracy by +10.26% on CIFAR10, +12.44% on CIFAR100 compared to InfoNCE. When scaled to a ViT-B/16 backbone, TeMo maintains these significant margins, surpassing InfoNCE by +10.99% on CIFAR10 and +9.09% on CIFAR100. While CWCL achieves higher accuracy in this specific setting, it does so at the cost of a sharp decline in retrieval performance (e.g., -7.38% on Flickr30k IR@1 vs. TeMo). In contrast, TeMo demonstrates superior multimodal consistency, delivering state-of-the-art retrieval results while remaining highly competitive on classification benchmarks.

In addition to the previously reported CC3M results, we also evaluate TeMo on ImageNet, where it achieves 34.79% Top-1 and 48.32% Top-3 accuracy, outperforming both InfoNCE and TS. We further compare TeMo with SoftCLIP on Top-1 accuracy, using the official results from the paper, which adopt the same pretraining dataset (CC3M) and ViT-B/16 architecture. Because differences in experimental environments and hyperparameters lead to slight variations in the CLIP baselines between our work and SoftCLIP, we focus on absolute performance gains: SoftCLIP achieves a +2.00 percentage points gain over its CLIP baseline, whereas our method attains a +2.23 percentage points gain when trained from scratch, Table 2. This indicates that TeMo is more effective at enhancing the semantic structure of embeddings than SoftCLIP under a shared backbone and pretraining regime. Extending pretraining to the larger CC12M dataset (Table 2), TeMo further improves to 41.76% Top-1 and 63.05% Top-3 accuracy, surpassing InfoNCE, TS and SLIP. These findings demonstrate that TeMo scales effectively with increasing dataset size, consistently delivering gains over strong baselines.

## 4.2 ABLATIONS

In Table 3, we ablate each component of the loss function in the Equation 8. In row a) we provide evaluation on a standard InfoNCE baseline.

**Multimodal Modulation.** In row b) we present evaluation of the standalone performance of the modulated multimodal $\mathcal{L}_{\text{M-MM}}$ loss without unimodal objectives and without scheduling, respectively. Compared to the InfoNCE baseline, we observe a significant drop in performance, suggesting that applying multimodal temperature modulation early in training introduces structural constraints that disrupt the learning of multimodal relationships. In particular, 3.32% drop in image retrieval (IR@1) and a 3.24% drop in text retrieval (TR@1) for MSCOCO. A similar trend is observed in the case of the retrieval performance for Flicker30k, with a drop of 5.96% for image retrieval (IR@1) and a decline of 8.60% for text retrieval (TR@1). We attribute this drop to the presence of hard negative samples that are semantically unrelated but highly similar to the anchor. Because the proposed temperature modulation assigns higher temperatures to more similar pairs, the pushing force

| Backbone | Method | MSCOCO | | Flickr30k | | CIFAR10 | | CIFAR100 | |
|---|---|---|---|---|---|---|---|---|---|
| | | IR@1 | TR@1 | IR@1 | TR@1 | Top-1 | Top-3 | Top-1 | Top-3 |
| RN50 | InfoNCE | 21.64 | 28.60 | 42.12 | 53.60 | 53.78 | 81.76 | 25.08 | 42.09 |
| | TS* Kukleva et al. (2023) | 22.01 | 28.20 | 42.90 | 53.30 | 51.61 | 82.03 | 28.53 | 45.21 |
| | DySTreSS* Manna et al. (2025) | 19.38 | 26.70 | 36.70 | 49.30 | 56.95 | 85.49 | 33.29 | 52.55 |
| | CWCL* Srinivasa et al. (2023) | 14.83 | 23.50 | 28.70 | 41.40 | 57.92 | 80.84 | 28.46 | 46.72 |
| | TeMo (ours) | **23.28** | **30.36** | **44.16** | **56.00** | 64.04 | 89.85 | 37.52 | 57.48 |
| ViT-B/16 | InfoNCE | 21.88 | **28.98** | 42.54 | 54.10 | 70.55 | 90.21 | 44.53 | 61.41 |
| | TS* Kukleva et al. (2023) | 21.45 | 27.32 | 41.02 | 49.90 | 72.02 | 91.65 | 47.13 | 65.45 |
| | DySTreSS* Manna et al. (2025) | 20.67 | 26.26 | 39.36 | 49.70 | 73.41 | 92.15 | 47.02 | 65.64 |
| | CWCL* Srinivasa et al. (2023) | 18.84 | 28.44 | 37.90 | 48.10 | **85.85** | **95.18** | **62.02** | **78.63** |
| | TeMo (ours) | **22.87** | 28.80 | **45.28** | **55.60** | 81.54 | 94.47 | 53.62 | 70.39 |

Table 1: **Zero-shot classification and crossmodal retrieval.** Models are trained on CC3M. Retrieval is evaluated on MSCOCO and Flickr30k using R@1 (%), with TR and IR denoting Text→Image and Image→Text. Zero-shot classification is reported on CIFAR10/100 with Top-1 and Top-3 accuracy (%). TS* indicates our multimodal adaptation of Temperature Schedules Kukleva et al. (2023). Similarly, DySTreSS* represent our multimodal adaptations of the method proposed by Manna et al. (2025), while CWCL* refers to our implementation of Srinivasa et al. (2023).

| Dataset | Method | Scr. | IN-1k | |
|---|---|---|---|---|
| | | | Top-1 | Top-3 |
| CC3M | InfoNCE (SoftCLIP)[†] | ✓ | 16.90 | – |
| | SoftCLIP[†] | ✓ | 18.90 | – |
| | InfoNCE | ✓ | 14.39 | 25.31 |
| | **TeMo (ours)** | ✓ | 16.62 | 27.97 |
| | InfoNCE | – | 28.00 | 40.23 |
| | TS* Kukleva et al. (2023) | – | 29.01 | 42.17 |
| | **TeMo (ours)** | – | **34.79** | **48.32** |
| CC12M | InfoNCE | – | 35.48 | 54.80 |
| | TS* Kukleva et al. (2023) | – | 38.90 | 59.02 |
| | SLIP Mu et al. (2021) | – | 39.77 | 59.42 |
| | **TeMo (ours)** | – | **41.76** | **63.05** |

Table 2: **Zero-shot classification on IN-1k.** Top-1 and Top-3 accuracy (%) of ViT-B/16 models pretrained on CC3M and CC12M using different configurations. Scr.: Trained from scratch. †: Results from original paper..

| | Base | Mod. | U.L | Sch. | MSCOCO | | Flickr30k | |
|---|---|---|---|---|---|---|---|---|
| | | | | | IR@1 | TR@1 | IR@1 | TR@1 |
| a) | ✓ | – | – | – | 21.64 | 28.60 | 42.12 | 53.60 |
| b) | – | ✓ | – | – | 18.32 | 25.36 | 36.16 | 45.00 |
| c) | ✓ | – | ✓(NM) | – | 20.65 | 25.44 | 39.16 | 50.10 |
| d) | ✓ | ✓ | – | – | 20.09 | 27.54 | 40.12 | 52.20 |
| e) | ✓ | ✓ | – | ✓ | 21.74 | 28.58 | 44.02 | 54.90 |
| f) | ✓ | ✓ | ✓ | – | 21.90 | 29.38 | 42.84 | 54.80 |
| g) | ✓ | ✓ | ✓ | ✓ | **23.28** | **30.36** | **44.16** | **56.00** |

Table 3: **Loss-component ablation.** Retrieval performance (%) under different training losses. Base: standard InfoNCE loss $\mathcal{L}_{MM}$; Mod.: modulated multimodal loss $\mathcal{L}_{M\text{-}MM}$; U. L.: additional unimodal losses, (NM denotes non-modulated objectives); Sch.: scheduler that blends $\mathcal{L}_{MM}$ with the temperature-modulated losses.

applied to hard negatives is significantly reduced. As a result, hard negatives remain close to the anchor, which limits the model's ability to learn effective separations at the beginning of the training.

**Impact of Standard Unimodal Losses.** To disentangle the benefits of unimodal supervision from our modulation scheme, we evaluate the baseline augmented with standard unimodal losses in row c). Surprisingly, simply adding unimodal objectives to the baseline degrades performance, dropping MSCOCO IR@1 from 21.64% (row a) to 20.65% (row c). While prior works have successfully combined InfoNCE with single unimodal objectives— LaCLIP Fan et al. (2023) utilizing language rewrites or SLIP Mu et al. (2021) via image supervision—integrating distinct losses for *both* modalities simultaneously appears to alter the alignment dynamics, potentially introducing optimization conflicts that require fine-grained tuning. In contrast, our full method (row g) achieves significant gains (23.28% IR@1), demonstrating that the proposed temperature modulation is essential for effectively leveraging unimodal objectives within this architecture.

**Multimodal Modulation and Scheduling.** Next, we examine the combination of the multimodal standard $\mathcal{L}_{MM}$ and modulated $\mathcal{L}_{M\text{-}MM}$ losses using the proposed progressive scheduling in line e). When the two losses are blended, performance generally improves over the $\mathcal{L}_{MM}$ (row a)) baseline. We argue that this gain is primarily driven by the dynamics introduced by the blending. At the early stages of training, it prioritizes the standard $\mathcal{L}_{MM}$ loss rather than the temperature-modulated counterpart. Due to the use of a low temperature in the $\mathcal{L}_{MM}$ loss, it applies a strong pushing force to all negatives, including the hard ones. As training progresses and the representation space becomes

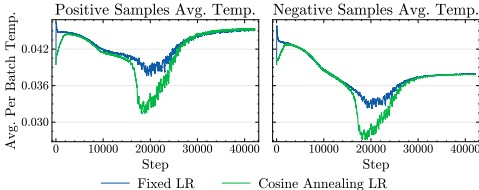
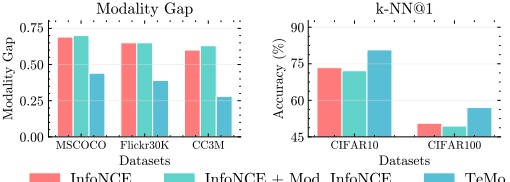

Figure 2: **Convergence of batch-wise temperatures.** Average batch-wise temperature slowly converges during training independent of the learning rate schedule.

Figure 3: **Modality gap and $k$-NN@1 evaluation.** Left: Modality gap on various datasets. Right: Visual-only representations. $k$-NN@1 accuracy on CIFAR10 and CIFAR100.

more structured, the influence of $\mathcal{L}_{\text{M-MM}}$ increases. The modulated loss reinforces the learned semantic structure, such as clustering similar instances, while continuing to refine the separation between negatives and anchors. This progressive transition allows the model to balance early-stage separation with later-stage semantic alignment.

**Unimodal Modulation.** Finally, in row g), we evaluate our full loss function, which combines the standard multimodal loss and modulated multimodal and unimodal losses with, being controlled via progressive scheduling. We observe substantial improvements over both the baseline (row a) vs. (row g)) and the variant using only scheduled multimodal modulation without unimodal losses (row e) vs. (row g)), resulting in the best overall performance. This highlights the crucial role of temperature-modulated unimodal objectives in learning strong representations.

**Importance of Progressive Scheduler.** To assess the impact of the blending mechanism, we fix the blending coefficients by setting $\alpha = 0.5$ and $\beta = 0.5$, effectively disabling the progressive blending between losses (rows (d) and (f)). Specifically, rows (d) and (e) use the same combination of multimodal losses, standard and modulated, but differ in their blending strategy: row (d) uses fixed coefficients, while row (e) employs a progressive scheduler. We observe consistent improvements across all metrics when the scheduler is used. Similarly, the full loss that includes unimodal components also benefits from progressive scheduling (row (f) vs. row (g)). These results suggest that using a lower temperature early in training helps capture fine-grained semantics, while broader semantic structures emerge later as the training progresses.

**Temperature Convergence.** To further understand the behavior of the proposed method, we analyze the temperature assignments during training. Specifically, we track the average temperature values for both positive and negative pairs on each batch, as shown in Figure 2. For both positive and negative pairs, we observe a drop at the early stages of the training, suggesting that the samples are overall less similar to each other. However, as the training progresses and the modulated losses take over, we observe a convergence of the average temperature used within a batch, for both the positives and negatives pairs. This phenomenon supports our hypothesis that the temperature-modulated losses reinforce the learned semantic structure and refine the separation between negatives and positives pairs. By the end of training, the temperatures assigned to positive pairs consistently converge to higher values than those assigned to negative pairs. To ensure that this pattern is not a result of the cosine annealing learning rate schedule, we conduct the same experiment with a fixed learning rate. Despite the learning rate scheduler, we arrive at the same result, indicating that convergence is not a result of it. We hypothesize this convergence is an indicator that the embedding space has reached a stable state.

**Modality Gap and $k$-NN Evaluation.** To further analyze the influence of the unimodal loss on both multimodal and unimodal representations, we measure the modality gap following Liang et al. (2022) and assess the geometry of the visual embedding space using $k$-NN evaluation (Figure 3). Specifically, we compare the standard multimodal InfoNCE baseline with variants that include standard and modulated multimodal losses, as well as our final TeMo loss that incorporates modulated unimodal objectives. We observe that the inclusion of unimodal losses leads to improved $k$-NN accuracy, indicating enhanced visual representations. In contrast, modulation of the multimodal loss alone has minimal impact on the visual embedding quality compared to standard InfoNCE. Interestingly, we observe the smallest modality gap when both multimodal and unimodal losses are

modulated, suggesting better alignment between modalities, an observation further supported by our ablation results in Table 3.

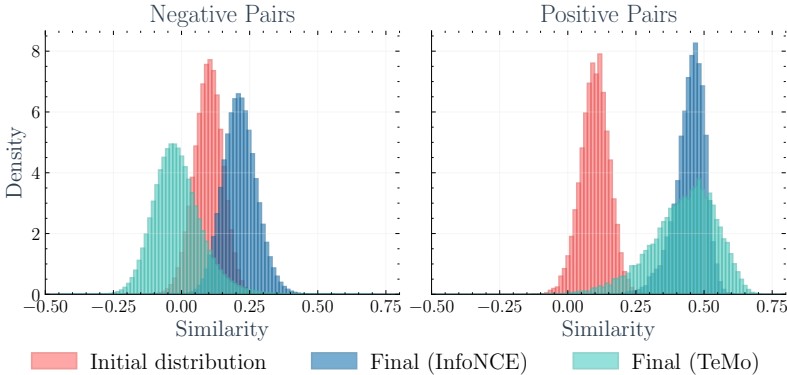

Figure 4: **Normalized distributions of image–text similarity on the CC3M validation set for negatives (left) and positive (right) pairs.** Red: the initial distribution represents before fine-tuning; blue: the distributions of negative and positive pairs after InfoNCE training; green: the distribution after training with TeMo loss.

**Distribution of Image-Text Similarity.** In Figure 4, we investigate the distribution of similarities between image and text embeddings for positive and negative pairs on the CC3M validation set. Before fine-tuning (red color), the distributions between the positive and negative pairs overlap considerably, indicating a poorly separated embedding space and that the model does not differentiate between similar and dissimilar samples. Fine-tuning with InfoNCE (blue color) shifts the distribution of the positive (similar) pairs to the right; however, we simultaneously observe that the distribution of negative pairs also shifts to higher similarities. As shown in the density histograms, this results in a significant overlap region where the similarity distributions of hard negatives and positives intersect. This overlap indicates a small or non-existent margin, leading to classification ambiguity. Similarly, TeMo shifts the distribution of positive pairs to the right. However, it simultaneously pushes the distribution of negatives in the opposite direction, effectively widening the gap between them and increasing the effective margin. This results in a cleaner and more discriminative embedding space. Notably, the distribution of the positive samples is wider, reflecting the model's ability to ignore false-positive pairs. For a more detailed discussion, please refer to the supplementary material, Appendix G. Consequently, representations learned with TeMo exhibit higher similarity among positive pairs and better separation from negatives. Overall, these results demonstrates that TeMo produces a significantly improved representation space, which directly contributes to stronger multimodal retrieval and zero-shot classification performance.

## 5 CONCLUSIONS

In this work, we proposed TeMo, a Temperature Modulation method for multimodal Contrastive Learning, which aims to improve the standard multimodal contrastive learning objective. First, we introduced an adaptive per-pair temperature modulation method, where the temperature for each training sample is adapted based on its local neighborhood in the embedding space. This allows the model to adjust the pushing force of each contrastive pair more effectively, adapting it to the respective neighborhood. Next, we extended this idea to the unimodal domain by integrating temperature modulation into the unimodal contrastive losses. These components are then combined into a unified modulated contrastive loss formulation, which further reinforces representation learning within each modality. Finally, we blended the modulated contrastive loss with the original InfoNCE objective with the progressive scheduling. This combination enables learning of instance-level details using a lower temperature, while our adaptive modulation framework gradually enables coarser semantic grouping. Our results demonstrated consistent improvement across datasets, MSCOCO, Flickr30K, CIFAR10, CIFAR100 and ImageNet-1k as well as evaluation metrics, highlighting the effectiveness of our method.

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

## A    APPENDIX

In the supplementary material, we provide extended versions of the tables from the main paper. Additionally, we include ablation studies that explore the use of expert-derived similarities, the effect of including unimodal losses from the beginning of training, and further insights into the robustness of our method against false positives.

## B    TEMO ALGORITHM

---

**Algorithm 1** TeMo algorithm

---

**Require:** $\mathbf{I}$ - batch of images; $\mathbf{T}$ - batch of texts; $\tilde{\mathbf{I}}$ - batch of augmented images; $\tilde{\mathbf{T}}$ - batch of augmented texts; $\mathcal{T}$ - fixed baseline temperature; $t$ - normalized training step ($t \in [0, 1]$); $\tau_{\mathbf{min}}$ - minimum temperature; $\tau_\alpha$ - temperature scaling factor;

1: **for** $(\mathbf{I}, \mathbf{T}, \tilde{\mathbf{I}}, \tilde{\mathbf{T}})$ in loader **do**
2:     **# Standard multimodal contrastive losses**
3:     $\mathcal{L}_{\text{I2T}} \leftarrow \mathcal{I}(\mathbf{I}, \mathbf{T}, \mathcal{T})$  # InfoNCE with const. $\tau$
4:     $\mathcal{L}_{\text{T2I}} \leftarrow \mathcal{I}(\mathbf{T}, \mathbf{I}, \mathcal{T})$  # InfoNCE with const. $\tau$
5:     $\mathcal{L}_{\text{MM}} \leftarrow \frac{1}{2}(\mathcal{L}_{\text{I2T}} + \mathcal{L}_{\text{T2I}})$
6:
7:     **# Modulated multimodal losses**
8:     $\mathcal{T}_{\text{I2T}} \leftarrow \lambda(\mathbf{I}, \mathbf{T})$  # modulated multimodal temp. mat.
9:     $\mathcal{T}_{\text{T2I}} \leftarrow \lambda(\mathbf{T}, \mathbf{I})$  # modulated multimodal temp. mat.
10:
11:     $\mathcal{L}_{\text{M-I2T}} \leftarrow \mathcal{I}(\mathbf{I}, \mathbf{T}, \mathcal{T}_{\text{I2T}})$  # InfoNCE with modul. $\tau$
12:     $\mathcal{L}_{\text{M-T2I}} \leftarrow \mathcal{I}(\mathbf{T}, \mathbf{I}, \mathcal{T}_{\text{T2I}})$  # InfoNCE with modul. $\tau$
13:     $\mathcal{L}_{\text{M-MM}} \leftarrow \frac{1}{2}(\mathcal{L}_{\text{MI2T}} + \mathcal{L}_{\text{MT2I}})$
14:
15:     **# Modulated unimodal losses**
16:     $\mathcal{T}_{\text{I2I}} \leftarrow \lambda(\mathbf{I}, \tilde{\mathbf{I}})$  # modulated unimodal temp. mat.
17:     $\mathcal{T}_{\text{T2T}} \leftarrow \lambda(\mathbf{T}, \tilde{\mathbf{T}})$  # modulated unimodal temp. mat.
18:
19:     $\mathcal{L}_{\text{M-I2I}} \leftarrow \mathcal{I}(\mathbf{I}, \tilde{\mathbf{I}}, \mathcal{T}_{\text{I2I}})$  # InfoNCE with modul. $\tau$
20:     $\mathcal{L}_{\text{M-T2T}} \leftarrow \mathcal{I}(\mathbf{T}, \tilde{\mathbf{T}}, \mathcal{T}_{\text{T2T}})$  # InfoNCE with modul. $\tau$
21:
22:     **# Compute loss weights (quadratic scheduler)**
23:     $\alpha \leftarrow (1 - t)^2$
24:     $\beta \leftarrow t^2$
25:
26:     **# Compute final TeMo loss**
27:     $\mathcal{L}_{\text{TeMo}} \leftarrow \alpha \cdot \mathcal{L}_{\text{MM}} + \beta \cdot (\mathcal{L}_{\text{M-MM}} + \mathcal{L}_{\text{M-I2I}} + \mathcal{L}_{\text{M-T2T}})$
28: **end for**

---

## C    COMPARISON TO STATE-OF-THE-ART

Tables 4 and  5 show a more detailed comparison with state-of-the-art models on zero-shot retrieval and zero-shot classification tasks. In addition to the metrics reported in the main paper, we also report R@5, R@10, and R-Mean (the average of the image-to-text and text-to-image) on MSCOCO and Flickr30k, as well as Top-3 and Top-5 accuracy on CIFAR-10, CIFAR-100, and ImageNet-1k. TeMo consistently delivers the strongest performance across every metric.

## D    ABLATIONS ON LOSS COMPONENTS

Table 6 presents the extended results of the ablation study in analyzing the contribution of each component in TeMo. Table 7 reports the same set of configurations, but evaluates them on zero-shot classification performance across 17 datasets from the LAION CLIP Benchmark. These results

| Backbone | Method | MSCOCO IR | | | MSCOCO TR | | | R-Mean | Flickr30k IR | | | Flickr30k TR | | | R-Mean |
|---|---|---|---|---|---|---|---|---|---|---|---|---|---|---|---|
| | | R@1 | R@5 | R@10 | R@1 | R@5 | R@10 | | R@1 | R@5 | R@10 | R@1 | R@5 | R@10 | |
| RN50 | InfoNCE | 21.64 | 45.77 | 57.47 | 28.60 | 53.90 | 66.56 | 45.66 | 42.12 | 68.84 | 78.76 | 53.60 | 81.30 | 88.90 | 68.92 |
| | TS* Kukleva et al. (2023) | 22.01 | 45.51 | 57.50 | 28.20 | 54.62 | 66.80 | 45.77 | 42.90 | 69.92 | 79.48 | 53.30 | 81.10 | 89.10 | 69.30 |
| | DySTreSS* Manna et al. (2025) | 19.38 | 42.34 | 54.27 | 26.70 | 52.24 | 64.02 | 43.16 | 36.70 | 64.98 | 75.66 | 49.30 | 77.80 | 87.10 | 65.26 |
| | DySTreSS Shifted* Manna et al. (2025) | 19.31 | 42.53 | 54.33 | 25.12 | 50.38 | 62.52 | 42.37 | 37.22 | 65.22 | 75.88 | 45.70 | 76.00 | 85.00 | 64.17 |
| | CWCL* ($\tau = 0.01$) Srinivasa et al. (2023) | 14.17 | 34.02 | 45.97 | 21.74 | 47.00 | 59.98 | 37.15 | 27.72 | 55.58 | 67.94 | 40.20 | 68.90 | 80.20 | 56.76 |
| | CWCL* ($\tau = 0.05$) Srinivasa et al. (2023) | 14.83 | 35.43 | 47.31 | 23.50 | 47.54 | 59.96 | 38.09 | 28.70 | 57.30 | 68.98 | 41.40 | 69.80 | 79.70 | 57.65 |
| | TeMo (ours) | **23.28** | **47.48** | **59.62** | **30.36** | **56.82** | 67.92 | **47.58** | **44.16** | **73.26** | **81.84** | **56.00** | **82.70** | **90.80** | **71.46** |
| ViT-B/16 | InfoNCE | 21.88 | 45.60 | 57.35 | **28.98** | **55.62** | 67.76 | 46.20 | 42.54 | 69.00 | 77.74 | 54.10 | 81.70 | 89.40 | 69.08 |
| | TS* Kukleva et al. (2023) | 21.45 | 44.31 | 55.61 | 27.32 | 53.14 | 65.88 | 44.62 | 41.02 | 68.86 | 78.68 | 49.90 | 80.90 | 88.60 | 67.99 |
| | DySTreSS* Manna et al. (2025) | 20.67 | 43.91 | 56.01 | 26.26 | 51.40 | 64.50 | 43.79 | 39.36 | 66.40 | 76.72 | 49.70 | 78.30 | 87.50 | 66.33 |
| | DySTreSS Shifted* Manna et al. (2025) | 21.33 | 44.80 | 56.82 | 25.40 | 52.20 | 64.66 | 44.20 | 40.74 | 68.64 | 78.90 | 49.90 | 77.40 | 85.50 | 66.85 |
| | CWCL* ($\tau = 0.01$) Srinivasa et al. (2023) | 17.90 | 39.87 | 52.02 | 26.12 | 51.94 | 64.98 | 42.14 | 36.62 | 64.98 | 74.86 | 49.00 | 78.40 | 86.90 | 65.13 |
| | CWCL* ($\tau = 0.05$) Srinivasa et al. (2023) | 18.84 | 41.12 | 53.59 | 28.44 | 54.48 | 66.80 | 43.88 | 37.90 | 67.06 | 77.56 | 48.10 | 79.00 | 88.20 | 66.30 |
| | TeMo (ours) | **22.87** | **46.66** | **58.88** | 28.80 | 55.48 | 67.66 | **46.73** | **45.28** | **73.14** | **81.58** | **55.60** | **82.60** | **89.50** | **71.28** |

Table 4: **Extension of Table 1 from the main paper**. Comparison with state-of-the-art retrieval performance (%) on the MSCOCO and Flickr30k datasets. DySTreSS* and DySTreSS Shifted* represent our multimodal adaptations of the method proposed by Manna et al. (2025), while CWCL* refers to our implementation of Srinivasa et al. (2023).

| Dataset | Backbone | Method | CIFAR10 | | | CIFAR100 | | | IN-1k | | |
|---|---|---|---|---|---|---|---|---|---|---|---|
| | | | Top-1 | Top-3 | Top-5 | Top-1 | Top-3 | Top-5 | Top-1 | Top-3 | Top-5 |
| CC3M | RN50 | InfoNCE | 53.78 | 81.76 | 90.38 | 25.08 | 42.09 | 50.40 | 30.70 | 43.31 | 48.34 |
| | | TS* Kukleva et al. (2023) | 51.61 | 82.03 | 92.08 | 28.53 | 45.21 | 52.57 | 31.86 | 45.25 | 50.38 |
| | | TeMo (ours) | **64.04** | **89.85** | **95.99** | **37.52** | **57.48** | **65.97** | **37.47** | **50.98** | **55.93** |
| | ViT-B/16 | InfoNCE | 70.55 | 90.21 | 95.50 | 44.53 | 61.41 | 68.13 | 28.00 | 40.23 | 45.15 |
| | | TS* Kukleva et al. (2023) | 72.02 | 91.65 | 96.03 | 47.13 | 65.45 | 72.17 | 29.01 | 42.17 | 47.20 |
| | | TeMo (ours) | **81.54** | **94.47** | **98.08** | **53.62** | **70.39** | **76.30** | **34.79** | **48.32** | **53.53** |
| CC12M | ViT-B/16 | InfoNCE | 65.18 | 89.40 | 96.22 | 37.12 | 57.34 | 65.42 | 35.48 | 54.80 | 62.54 |
| | | TS* Kukleva et al. (2023) | 71.31 | 91.28 | 96.31 | 39.29 | 60.69 | 69.60 | 38.91 | 59.02 | 67.27 |
| | | SLIP Mu et al. (2021) | 68.50 | 89.27 | 94.59 | 45.08 | 66.38 | **74.57** | 39.77 | 63.05 | 67.08 |
| | | TeMo (ours) | **78.63** | **92.42** | **96.42** | **45.24** | **66.49** | 74.36 | **41.76** | **63.05** | **71.41** |

Table 5: **Extension of Table 1 and 2 from the main paper**. Comparison of zero-shot Top-K classification accuracy (%) on CIFAR10, CIFAR100 and ImageNet-1k where K $\in \{1, 3, 5\}$.

demonstrate that our proposed method achieves consistent and significant improvements over the baseline across most tasks.

| Base | Mod. | U.L. | Sch. | MSCOCO IR | | | MSCOCO TR | | | R-Mean | Flickr30k IR | | | Flickr30k TR | | | R-Mean |
|---|---|---|---|---|---|---|---|---|---|---|---|---|---|---|---|---|---|
| | | | | R@1 | R@5 | R@10 | R@1 | R@5 | R@10 | | R@1 | R@5 | R@10 | R@1 | R@5 | R@10 | |
| ✔ | – | – | – | 21.64 | 45.77 | 57.47 | 28.60 | 53.90 | 66.56 | 45.66 | 42.12 | 68.84 | 78.76 | 53.60 | 81.30 | 88.90 | 68.92 |
| – | ✔ | – | – | 18.32 | 40.85 | 52.79 | 25.36 | 50.60 | 62.70 | 41.77 | 36.16 | 65.16 | 74.78 | 45.00 | 75.60 | 84.70 | 63.57 |
| ✔ | – | ✔ (NM) | – | 20.65 | 43.81 | 55.85 | 25.44 | 51.86 | 64.02 | 43.61 | 39.16 | 67.46 | 78.06 | 50.10 | 78.00 | 86.30 | 66.51 |
| ✔ | ✔ | – | – | 20.09 | 44.45 | 56.55 | 27.54 | 53.40 | 65.22 | 44.68 | 40.12 | 67.32 | 77.24 | 52.20 | 80.90 | 87.70 | 67.58 |
| ✔ | ✔ | – | ✔ | 22.44 | 46.06 | 58.07 | 29.92 | **57.00** | **67.98** | 46.91 | 43.24 | 70.56 | 79.40 | 54.50 | 82.50 | 89.80 | 70.00 |
| ✔ | ✔ | ✔ | – | 21.90 | 46.10 | 58.25 | 29.38 | 56.54 | 68.62 | 46.80 | 42.84 | 71.70 | 80.94 | 54.80 | 80.80 | 89.20 | 70.05 |
| ✔ | ✔ | ✔ | ✔ | **23.28** | **47.48** | **59.62** | **30.36** | 56.82 | 67.92 | **47.58** | **44.16** | **73.26** | **81.84** | **56.00** | **82.70** | **90.80** | **71.46** |

Table 6: **Extension of Table 3 from the main paper**. Comparison of retrieval performance (%) on the MSCOCO and Flickr30k datasets for different training configurations. (NM denotes non-modulated objectives.)

# E $k$-NN Evaluation on Visual Representations

Table 8 shows extended $k$-NN evaluation results on visual representations, providing a tabular version of Figure 3 (right). In addition to $k$-NN@1, we additionally report $k$-NN@10. Similar to the main paper, we observe that the inclusion of unimodal losses plays a crucial role in $k$-NN accuracy, enabling TeMo to outperform other baselines.

# F Mixture of Experts

We further investigate the effect of the similarity matrix on the temperature-modulated losses. We employ the DINOv2-Small Oquab et al. (2023) model to extract patch-level features, which we

| Base | Mod. | U.L. | Sch. | Zero-shot Classification (Top-1 Acc %) | | | | | | | | | | | | | | | | |
|---|---|---|---|---|---|---|---|---|---|---|---|---|---|---|---|---|---|---|---|---|
| | | | | C10 | C100 | C211 | DTD | ESAT | GTSRB | FGVCA | IN-S | IN-A | IN-R | IN-O | FLO | PETS | PCAM | MNIST | F101 | IN1k |
| ✔ | – | – | – | 53.78 | 25.08 | 1.28 | 24.20 | 17.34 | 8.11 | 1.14 | 19.19 | 7.12 | 39.01 | 37.20 | 14.34 | 21.32 | 49.84 | **13.30** | 20.29 | 30.70 |
| – | ✔ | – | – | 50.85 | 29.08 | 1.08 | 19.41 | 16.04 | 7.73 | 0.87 | 19.18 | 8.16 | 38.78 | 36.95 | 12.31 | 16.89 | 50.69 | 11.50 | 20.61 | 31.10 |
| ✔ | – | ✔ (NM) | – | 61.96 | 34.22 | 1.58 | 23.35 | **29.73** | 6.48 | 1.05 | 21.11 | 7.89 | 39.68 | 41.70 | 11.46 | 17.71 | 57.24 | 6.03 | 17.74 | 36.74 |
| ✔ | ✔ | – | – | 51.64 | 29.06 | 1.10 | 21.75 | 12.38 | 8.14 | 1.29 | 19.90 | 7.69 | 39.18 | 39.25 | 14.57 | 20.85 | 52.29 | 10.34 | 19.18 | 31.69 |
| ✔ | ✔ | – | ✔ | 49.28 | 26.27 | 1.34 | 23.56 | 22.90 | 8.76 | 1.17 | 20.91 | 8.45 | 42.01 | 39.70 | **14.77** | 21.83 | 54.94 | 10.01 | 20.67 | 32.94 |
| ✔ | ✔ | ✔ | – | 60.92 | 34.90 | 1.54 | 26.22 | 16.26 | **9.60** | **1.71** | 24.17 | 9.19 | 44.01 | 45.40 | 12.93 | 18.42 | **61.81** | 8.61 | 20.99 | 37.04 |
| ✔ | ✔ | ✔ | ✔ | **64.04** | **37.52** | **1.68** | **27.29** | 25.61 | 9.14 | 1.08 | **25.17** | **9.81** | **44.42** | **46.55** | 13.04 | **21.91** | 61.51 | 10.50 | **22.00** | **37.47** |

Table 7: **Zero-shot Top-1 classification accuracy (%) across datasets.** C10 = CIFAR10, C100 = CIFAR100, C211 = Country211, DTD = Describable Textures Dataset, ESAT = EuroSAT, GT-SRB = German Traffic Sign Recognition Benchmark, FGVCA = FGVC-Aircraft, IN-S = ImageNet-Sketch, IN-A = ImageNet-A, IN-R = ImageNet-R, IN-O = ImageNet-O, FLO = Oxford Flowers-102, PETS = Oxford-IIIT Pets, PCAM = PatchCamelyon, MNIST = MNIST, F101 = Food-101, IN1k = ImageNet-1k. Base = standard InfoNCE loss $\mathcal{L}_{MM}$; Mod. = modulated multimodal loss $\mathcal{L}_{M\text{-}MM}$; U.L. = additional unimodal losses; Sch. = temperature scheduling; NM denotes non-modulated objectives.)

| Base | Mod. | U.L. | Sch. | CIFAR10 | | CIFAR100 | |
|---|---|---|---|---|---|---|---|
| | | | | $k$-NN@1 | $k$-NN@10 | $k$-NN@1 | $k$-NN@10 |
| ✔ | – | – | – | 73.40 | 77.29 | 50.59 | 53.27 |
| – | ✔ | – | – | 68.66 | 72.98 | 44.72 | 48.61 |
| ✔ | – | ✔ (NM) | – | 80.69 | 83.88 | 56.40 | 60.83 |
| ✔ | ✔ | – | – | 73.43 | 77.43 | 50.36 | 53.87 |
| ✔ | ✔ | – | ✔ | 72.14 | 76.13 | 49.44 | 52.55 |
| ✔ | ✔ | ✔ | – | 80.27 | 83.69 | 56.18 | 59.68 |
| ✔ | ✔ | ✔ | ✔ | **80.70** | **83.89** | **57.04** | **60.65** |

Table 8: **Extended version and tabular representation of Figure 3 (Right).** $k$-NN evaluation results on visual representations. Metrics shown are $k$-NN@1 and $k$-NN@10 accuracy (%) for CIFAR10 and CIFAR100. ✔ indicates the presence of Base loss, Modality interaction, Unimodal losses (U.L.), Scheduling (Sch.). NM denotes non-modulated objectives.

then averaged to produce a single global image embedding per each sample. These embeddings are used to compute image-to-image similarity matrices for temperature modulation. For text, we use the pre-trained model all-roberta-large-v1 Reimers & Gurevych (2019) to generate sentence embeddings, which serve as the basis for computing text-to-text similarity matrices.

First, we modulate the temperature using only one type of expert-derived similarity—either image-to-image (I2I) or text-to-text (T2T). Next, we incorporate both experts simultaneously. The results shown in Tables 9 and 10 indicate that the use of expert-based similarity matrices improves the performance on most metrics compared to the InfoNCE baseline. However, even with these gains, the expert-based configurations are consistently outperformed by our proposed method. Notably, TeMo achieves better performance without relying on any external models or adding additional computational overhead, highlighting its effectiveness and efficiency as a standalone approach.

| Method | MSCOCO IR | | | MSCOCO TR | | | R-Mean | Flickr30k IR | | | Flickr30k TR | | | R-Mean |
|---|---|---|---|---|---|---|---|---|---|---|---|---|---|---|
| | R@1 | R@5 | R@10 | R@1 | R@5 | R@10 | | R@1 | R@5 | R@10 | R@1 | R@5 | R@10 | |
| InfoNCE | 21.64 | 45.77 | 57.47 | 28.60 | 53.90 | 66.56 | 45.66 | 42.12 | 68.84 | 78.76 | 53.60 | 81.30 | 88.90 | 68.92 |
| Text Expert | 22.01 | 45.84 | 57.63 | 28.76 | 55.48 | 67.22 | 46.16 | 42.46 | 69.64 | 79.12 | 53.10 | 81.50 | 88.90 | 69.12 |
| Vision Expert | 21.87 | 45.47 | 57.30 | 28.60 | 56.00 | 67.38 | 46.10 | 41.58 | 68.88 | 78.24 | 53.20 | 80.30 | 87.40 | 68.71 |
| Text & Vision Expert | 21.93 | 45.55 | 57.52 | 29.56 | 55.96 | 67.46 | 46.33 | 41.00 | 69.12 | 78.82 | 54.10 | 80.50 | 89.00 | 68.75 |
| TeMo (ours) | **23.28** | **47.48** | **59.62** | **30.36** | **56.82** | **67.92** | **47.58** | **44.16** | **73.26** | **81.84** | **56.00** | **82.70** | **90.80** | **71.46** |

Table 9: **Retrieval results with expert-informed similarities.** Retrieval performance (%) on MSCOCO and Flickr30k using expert-informed similarity matrices for temperature modulation.

| Method | CIFAR10 | | | CIFAR100 | | |
|---|---|---|---|---|---|---|
| | Top-1 | Top-3 | Top-5 | Top-1 | Top-3 | Top-5 |
| InfoNCE | 53.78 | 81.76 | 90.38 | 25.08 | 42.09 | 50.40 |
| Text Expert | 51.09 | 79.45 | 88.51 | 24.56 | 41.00 | 49.67 |
| Vision Expert | 54.12 | 83.90 | 93.67 | 30.09 | 47.16 | 56.01 |
| Text & Vision Expert | 54.95 | 83.00 | 93.61 | 26.77 | 43.45 | 52.00 |
| TeMo (ours) | **64.04** | **89.85** | **95.99** | **37.52** | **57.48** | **65.97** |

Table 10: **Comparison of zero-shot classification accuracy.** Zero-shot Top-K accuracy (%) on CIFAR10 and CIFAR100 using expert-informed similarity matrices for temperature modulation. TeMo achieves the best results across all metrics.

## G    ROBUSTNESS AGAINST FALSE-POSITIVES

Figure 5 and 6 illustrate the robustness of the proposed method against false positives. Although the image-caption pairs are treated as positives due to their dataset annotations, our approach learns to distinguish mismatched pairs. On the far left of the distribution Figure 5, we observe examples where the caption does not match the image—these are actually false positive pairs. In contrast, pairs on the right side of the distribution exhibit strong semantic alignment, which our method correctly identifies. Notably, the standard deviation of the positive pair scores under TeMo is higher than that of the InfoNCE baseline, suggesting that TeMo reinforces truly meaningful associations while mitigating noise from misleading labels.

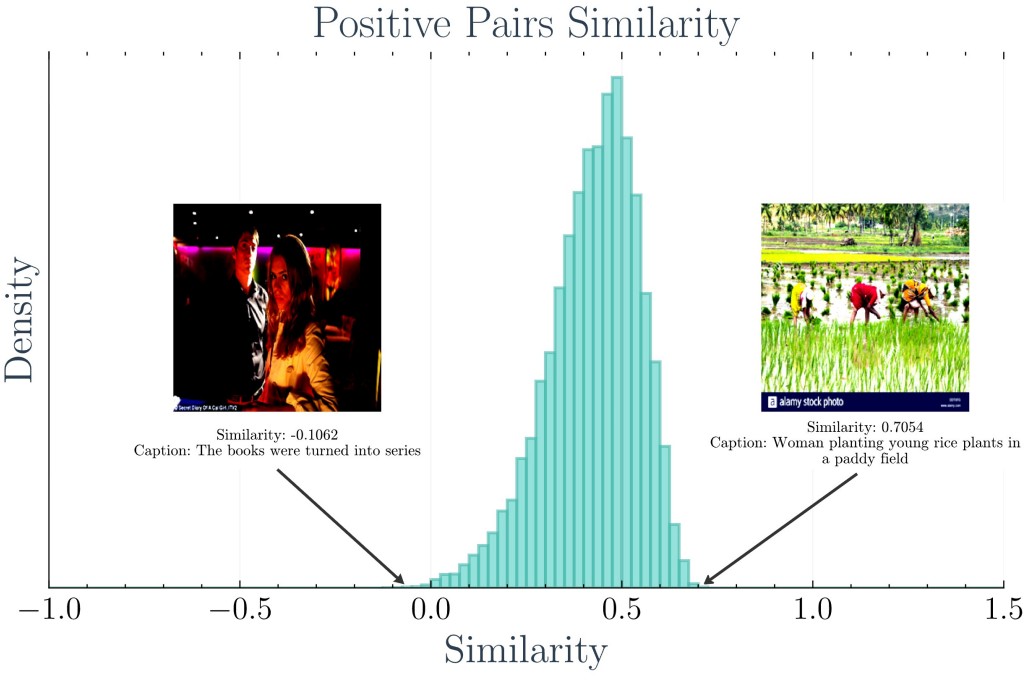

Figure 5: **Similarity distribution of positive pairs**. TeMo effectively separates semantically aligned pairs from mismatched one. Additional examples are provided in Figure 6.

## H    LIMITATIONS

Our experiments are constrained by both batch size and pretraining dataset scale. Due to limited resources, we train with a maximum batch size of 4096 and only pretrain on CC3M and CC12M, with the latter being the largest dataset we use. In contrast, many recent works rely on significantly

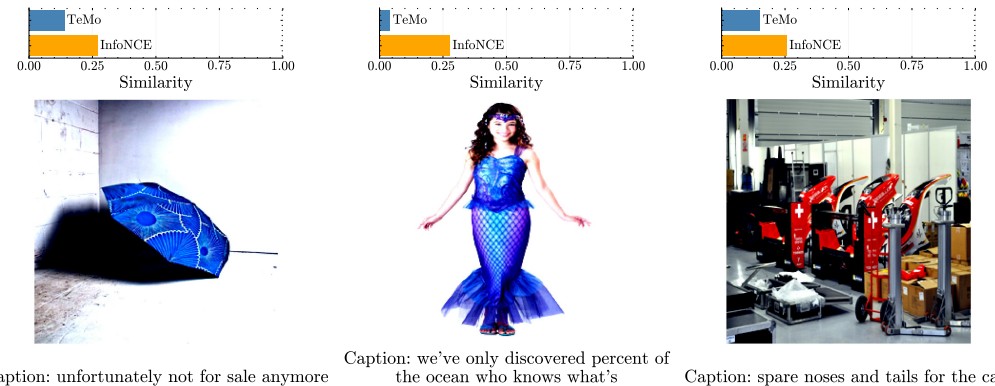

Caption: unfortunately not for sale anymore

Caption: we've only discovered percent of the ocean who knows what's in the rest of it

Caption: spare noses and tails for the cars

Figure 6: **Examples of false-positive image–caption pairs.** While InfoNCE assigns relatively high similarity scores to these mismatched examples, TeMo assigns consistently lower scores, indicating improved discrimination and robustness to label noise.

larger batch sizes (e.g., LaCLIP and LaSLIP Fan et al. (2023) at 8192, and large-scale models such as EvaCLIP Sun et al. (2023), SigLIP Zhai et al. (2023), SigLIP2 Tschannen et al. (2025), and TULIP Tang et al. with 30k–170k) as well as much larger and more diverse pretraining datasets (e.g., YFCC15M Gu et al. (2024), LAION Schuhmann et al. (2022), or merged multi-billion scale dataset). Since performance is highly sensitive to both batch size and data scale, our results with TeMo should be viewed as competitive under constrained settings, rather than directly comparable to large-batch, large-data methods.

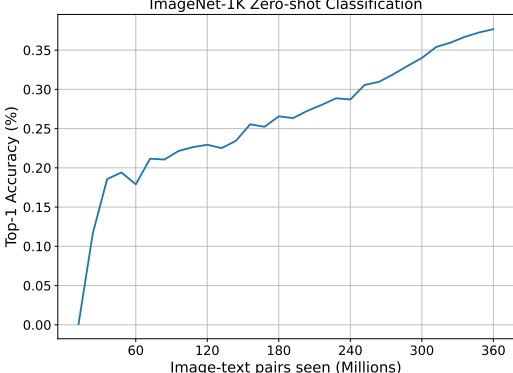

Figure 7: **ImageNet-1K zero-shot accuracy throughout training.** We report the Top-1 accuracy as a function of the cumulative number of image-text pairs seen (in millions). The model is trained on CC12M for 30 epochs, resulting in a total of 360M samples seen.

However, when viewed from the perspective of the number of image-text samples seen during training, TeMo remains competitive with methods such as CWCL Srinivasa et al. (2023) or LiT Zhai et al. (2021). Training on CC12M for 30 epochs results in a cumulative 360M samples seen, confirming that TeMo performs effectively at a computational scale comparable to established methods, as illustrated in Figure 7.

## I  CONNECTING TEMO TO EXISTING THEORETICAL FRAMEWORKS

TeMo can be viewed as a dynamic mechanism that balances Uniformity and Alignment by adaptively modulating the temperature $\tau$. Building on Wang & Liu (2021), we set lower $\tau$ for dissimilar pairs to sharpen gradients that repel samples and promote global Uniformity, while increasing $\tau$ for high-similarity pairs to relax penalties on hard positives and emphasize semantic Alignment.

Training proceeds with a coarse-to-fine scheduler: in Phase 1, $\tau$ is fixed and low to expand the embedding space and avoid collapse; in Phase 2, per-instance modulation is activated to refine local neighborhoods and consolidate semantic clusters formed earlier. This instance-level control prevents "over-separation" of related concepts during later stages while still maintaining a well-spread representation. Ablations corroborate this schedule: applying strong alignment too early—before adequate global Uniformity is established—reduces performance (Table 3, row b), supporting the necessity of the temporal progression from Uniformity to Alignment.

## J    IMPACT OF TEMO ON THE SMOOTHNESS OF THE OBJECTIVE

The proposed modulation acts as a data-dependent regularizer. In standard InfoNCE with a fixed low temperature, high-similarity negative pairs produce extremely sharp gradients, creating "cliffs" in the loss landscape that can destabilize training. Our method increases the temperature as similarity increases. This locally "softens" the softmax distribution for these high-similarity pairs, effectively dampening the gradient magnitude and preventing explosive updates from hard negatives.

## K    BATCH SIZE STANDARDIZATION

To ensure a strictly fair and consistent comparison, we standardize the batch size across all additional experiments. Specifically, we evaluate methods using a batch size of 2048 to match the configuration of TeMo. We verified that this change does not alter the conclusions; the relative performance trends observed at 1024 are consistent with preliminary experiments conducted at a larger batch size of 2048.

## L    LLM USAGE

A large language model was used to improve the clarity and the readability of the paper. All edits were reviewed and approved by the authors.

