# OpenReview forum: "TeMo: Temperature Modulation for Multimodal Contrastive Learning"
_ICLR.cc/2026/Conference — Submitted to ICLR 2026_

### Official Review · Reviewer_sR36 · 2025-11-01

**Soundness:** 1
**Presentation:** 3
**Contribution:** 2
**Rating:** 2
**Confidence:** 4

**Summary:**

This paper introduces a dynamic temperature for the InfoNCE loss, calculated on a per-element-pair basis. The core idea is that the temperature value should be a function of the pair's embedding similarity. The paper's strategy is to assign pairs with high similarity a correspondingly higher temperature, with the goal of more appropriately scaling the loss contribution from different positive and negative pairs.

**Strengths:**

The primary strength of the work is its simplicity. The proposed temperature modulation technique appears straightforward to implement and is presented as a general method that can be integrated with various existing models to potentially improve performance.

**Weaknesses:**

**Unclear Attribution of Performance Gains**: The central weakness of the paper is the ambiguity in attributing the reported performance gains specifically to the novel temperature modulation. The main results (e.g., in Tables 1 & 2) appear to combine the proposed modulation with additional Unimodal losses. Adding a unimodal loss is a known technique for improving performance in this domain.

To properly isolate the benefit of the proposed method, a crucial baseline is missing from the comparisons: the performance of the base model with the Unimodal loss but without the proposed temperature modulation. Without this direct comparison, it is impossible to determine whether the reported gains stem from the novel modulation or simply from the addition of the Unimodal loss.

This concern is further amplified by the ablation study, which shows that the 'Base + MM + Scheduling' method yields mixed and marginal gains compared to the 'TS*' baseline. For example, the proposed method achieves 21.74 (MS IR @1), 28.58 (MS TR @1), 44.02 (FL IR@1), and 54.90 (FL TR@1), whereas the baseline TS* achieves 22.01 (MS IR @1), 28.20 (MS TR @1), 42.90 (FL IR@1), and 53.30 (FL TR@1) , respectively. These indicate only a marginal improvement.

**Insufficient Zero-Shot Evaluation**: The empirical evaluation for zero-shot classification is insufficient. The results are limited to CIFAR-10 and CIFAR-100, which are not representative. Current practice for evaluating CLIP-like models involves reporting metrics on more challenging suites, such as those included in ELEVATER [1] (e.g., Image Classification-in-the-Wild, which includes 20 zero-shot classification tasks) and other common ImageNet variants (e.g., ImageNet-Adversarial, ImageNet-R, and ImageNet-V2). The absence of these results makes it difficult to assess the method's practical utility.

**Lack of Comparison to Hard Negative Mining**: The proposed mechanism, increasing the temperature for high-similarity pairs, effectively alters the gradient contribution of these samples. This approach is conceptually related to various hard negative mining strategies, which also focus training on the most informative (often most similar) negative pairs. The paper fails to discuss this connection or provide any empirical comparison against established hard negative mining techniques. This comparison is necessary to position the contribution relative to existing literature and to determine if this dynamic temperature formulation offers tangible advantages over prior methods.

[1] Li, Chunyuan, et al. "Elevater: A benchmark and toolkit for evaluating language-augmented visual models." Advances in Neural Information Processing Systems 35 (2022): 9287-9301.

**Questions:**

See weaknesses

---

> ### Author Response · Authors · 2025-11-23
>
> We thank the reviewer for the time taken to provide detailed and constructive feedback to our work. We appreciate that the reviewer acknowledges the simplicity of our framework and that it “can be integrated with various existing models to potentially improve performance.”
>
> In the following, we would like to address the reviewer's remaining concerns.
>
> > **W1. Unclear Attribution of Performance Gains**
>
> We thank the reviewer for identifying this gap in our ablation study. We agree that isolating the effect of the Unimodal Loss (U.L.) from our proposed Temperature Modulation is crucial to verify the source of the performance gains.
>
> To address this, we conducted additional experiments, specifically evaluating the "Base + Unimodal Loss" configuration without temperature modulation, as requested. In the table below, we extend our Table 3 with these additional ablations.
>
> | Base | Mod. | U.L. | Sch. | COCO TR - R@1 | COCO TR - R@5 | COCO TR - R@10 | COCO IR - R@1 | COCO IR - R@5 | COCO IR - R@10 | R-Mean | Flickr TR - R@1 | Flickr TR - R@5 | Flickr TR - R@10 | Flickr IR - R@1 | Flickr IR - R@5 | Flickr IR - R@10 | R-Mean |
> | :---: | :---: | :---: | :---: | :---: | :---: | :---: | :---: | :---: | :---: | :---: | :---: | :---: | :---: | :---: | :---: | :---: | :---: |
> | ✓ | - | - | - | 21.64 | 45.77 | 57.47 | 28.60 | 53.90 | 66.56 | 45.66 | 42.12 | 68.84 | 78.76 | 53.60 | 81.30 | 88.90 | 68.92 |
> | - | ✓ | - | - | 18.32 | 40.85 | 52.79 | 25.36 | 50.60 | 62.70 | 41.77 | 36.16 | 65.16 | 74.78 | 45.00 | 75.60 | 84.70 | 63.57 |
> | ✓ | - | ✓ (No modulation) | - | 19.99 | 43.13 | 55.30 | 26.24 | 52.42 | 64.20 | 43.55 | 38.72 | 68.62 | 77.66 | 48.90 | 76.90 | 86.60 | 66.23 |
> | ✓ | ✓ | - | - | 20.09 | 44.45 | 56.55 | 27.54 | 53.40 | 65.22 | 44.68 | 40.12 | 67.32 | 77.24 | 52.20 | 80.90 | 87.70 | 67.58 |
> | ✓ | ✓ | - | ✓ | 22.44 | 46.06 | 58.07 | 29.92 | **57.00** | **67.98** | 46.91 | 43.24 | 70.56 | 79.40 | 54.50 | 82.50 | 89.80 | 70.00 |
> | ✓ | ✓ | ✓ | - | 21.90 | 46.10 | 58.25 | 29.38 | 56.54 | 68.62 | 46.80 | 42.84 | 71.70 | 80.94 | 54.80 | 80.80 | 89.20 | 70.05 |
> | ✓ | ✓ | ✓ | ✓ | **23.28** | **47.48** | **59.62** | **30.36** | 56.82 | 67.92 | **47.58** | **44.16** | **73.26** | **81.84** | **56.00** | **82.70** | **90.80** | **71.46** |
>
> Analysis of the New Baseline: The results from our experiments (3rd row) does not support the hypothesis that the U.L is the primary driver of performance. Adding the U.L without our modulation results in an R-Mean of 43.55% on MSCOCO (Row 3).
>
> While prior works have successfully combined the standard InfoNCE loss with single unimodal objectives—such as Fan et al. [1] utilizing language rewrites, or Mu et al. [2] adding image supervision via a SimCLR [3] loss—we are not aware of work that combines both text and vision unimodal losses and provides thorough ablations on that specific integration. Consequently, we assume that including distinct losses for both modalities changes the dynamics of alignment and requires further fine-grained hyperparameter tuning. On the other hand, our method (Last Row), which combines U.L. with the proposed Temperature Modulation, achieves an R-Mean of 47.58%.
>
> These findings demonstrate that the U.L alone is insufficient for this architecture and can even be detrimental. The performance gains are, therefore, attributable to the proposed Temperature Modulation.
>
> [1] L. Fan, D. Krishnan, P. Isola, D. Katabi, and Y. Tian, “Improving CLIP Training with Language Rewrites,” Oct. 28, 2023, arXiv: arXiv:2305.20088. doi: 10.48550/arXiv.2305.20088.
>
> [2] N. Mu, A. Kirillov, D. Wagner, and S. Xie, “SLIP: Self-supervision meets Language-Image Pre-training,” Dec. 23, 2021, arXiv: arXiv:2112.12750. doi: 10.48550/arXiv.2112.12750.
>
> [3] T. Chen, S. Kornblith, M. Norouzi, and G. Hinton, “A Simple Framework for Contrastive Learning of Visual Representations,” July 01, 2020, arXiv: arXiv:2002.05709. doi: 10.48550/arXiv.2002.05709.

---

> > ### Author Response · Authors · 2025-11-23
> >
> > > **W2. Insufficient Zero-Shot Evaluation**
> >
> > We thank the reviewer for suggesting another benchmark to evaluate our model. We agree that limiting the evaluation to C10, C100 and IN-1k provides an incomplete picture of the model's generalization capabilities. To address this concern, we have conducted a comprehensive new evaluation on 17 distinct zero-shot classification benchmarks using CLIP_Benchmark from the LAION team.
> >
> > ### Zero-shot Top-1 Classification Accuracy (%)
> > | Base | Mod. | U.L. | Sch. | C10 | C100 | C211 | DTD | ESAT | GTSRB | FGVCA | IN-S | IN-A | IN-R | IN-O | FLO | PETS | PCAM | MNIST | F101 | IN1k |
> > | :---: | :---: | :---: | :---: | :---: | :---: | :---: | :---: | :---: | :---: | :---: | :---: | :---: | :---: | :---: | :---: | :---: | :---: | :---: | :---: | :---: |
> > | ✓ | - | - | - | 53.78 | 25.08 | 1.28 | 24.20 | 17.34 | 8.11 | 1.14 | 19.19 | 7.12 | 39.01 | 37.20 | 14.34 | 21.32 | 49.84 | **13.30** | 20.29 | 30.70 |
> > | - | ✓ | - | - | 50.85 | 29.08 | 1.08 | 19.41 | 16.04 | 7.73 | 0.87 | 19.18 | 8.16 | 38.78 | 36.95 | 12.31 | 16.89 | 50.69 | 11.50 | 20.61 | 31.10 |
> > | ✓ | - | ✓ (No modulation) | - | 62.67 | 32.30 | 1.56 | 20.70 | **28.17** | **10.60** | 0.63 | 19.66 | 7.62 | 37.61 | 37.15 | 11.31 | **24.01** | **66.70** | 11.67 | 16.58 | 35.72 |
> > | ✓ | ✓ | - | - | 51.64 | 29.06 | 1.10 | 21.75 | 12.38 | 8.14 | 1.29 | 19.90 | 7.69 | 39.18 | 39.25 | 14.57 | 20.85 | 52.29 | 10.34 | 19.18 | 31.69 |
> > | ✓ | ✓ | - | ✓ | 49.28 | 26.27 | 1.34 | 23.56 | 22.90 | 8.76 | 1.17 | 20.91 | 8.45 | 42.01 | 39.70 | **14.77** | 21.83 | 54.94 | 10.01 | 20.67 | 32.94 |
> > | ✓ | ✓ | ✓ | - | 60.92 | 34.90 | 1.54 | 26.22 | 16.26 | 9.60 | **1.71** | 24.17 | 9.19 | 44.01 | 45.40 | 12.93 | 18.42 | 61.81 | 8.61 | 20.99 | 37.04 |
> > | ✓ | ✓ | ✓ | ✓ | **64.04** | **37.52** | **1.68** | **27.29** | 25.61 | 9.14 | 1.08 | **25.17** | **9.81** | **44.42** | **46.55** | 13.04 | 21.91 | 61.51 | 10.50 | **22.00** | **37.47** |
> > ---
> > **Legend & Definitions**
> > * **Components:**
> >     * **Base:** standard InfoNCE loss $\mathcal{L}_{\text{MM}}$
> >     * **Mod.:** modulated multimodal loss $\mathcal{L}_{\text{M-MM}}$
> >     * **U.L.:** additional unimodal losses
> >     * **Sch.:** scheduler that blends $\mathcal{L}_{\text{MM}}$ with the temperature-modulated losses
> > * **Datasets:**
> >     * **C10/C100:** CIFAR10 / CIFAR100
> >     * **C211:** Country211
> >     * **DTD:** Describable Textures Dataset
> >     * **ESAT:** EuroSAT
> >     * **GTSRB:** German Traffic Sign Recognition Benchmark
> >     * **FGVCA:** FGVC-Aircraft
> >     * **IN-S/A/R/O:** ImageNet-Sketch / ImageNet-A / ImageNet-R / ImageNet-O
> >     * **FLO:** Oxford Flowers-102
> >     * **PETS:** Oxford-IIIT Pets
> >     * **PCAM:** PatchCamelyon
> >     * **F101:** Food-101
> >     * **IN1k:** ImageNet-1k
> >
> > As presented in the table above, our evaluation now includes ImageNet variants as well as fine-grained and diverse datasets.
> > The results demonstrate that our proposed method achieves consistent and significant improvements over the baseline across most tasks. Interestingly, the additional experiment combining unimodal and multimodal losses performs best on datasets such as ESTAT, GTSRB, PETS, and PCAM. These datasets share common characteristics: their semantics are primarily visually discriminative, while their text labels lack rich vision–language co-occurrence. We therefore hypothesize that these datasets benefit more from the inclusion of unimodal losses, which enhance the representation quality of each modality independently.
> >
> > > **W3. Lack of Comparison to Hard Negative Mining**
> >
> > We appreciate the reviewer’s insight regarding the conceptual relationship between our proposed dynamic temperature mechanism and Hard Negative Mining (HNM) strategies.
> >
> > In the revised manuscript, we will explicitly discuss this connection, positioning our work alongside methods such as SoftCLIP to clarify how our dynamic temperature formulation compares to existing strategies. Regarding the empirical comparison, while we note that some of our current baselines already employ standard hard negative strategies, we want to ensure we address the reviewer's specific expectations. To that end, we would be grateful if the reviewer could provide references to the specific "established hard negative mining techniques" they believe would provide the most valuable comparison.

---

> > > ### Comment · Reviewer_sR36 · 2025-11-27
> > >
> > > I thank the authors for the detailed response.
> > >
> > > 1. You have resolved my issue about the Unimodal losses.
> > > 2. Thank you for the additional zero-shot results.
> > > 3. Regarding hard mining, to clarify, I was hoping you would be able to add a literature discussion in the paper showing potential connections. I did not mean an empirical comparison. Hope that clarifies that point.
> > >
> > > I have updated my scores to reflect the new information in your responses.

---

> > > > ### Author Response · Authors · 2025-12-03
> > > >
> > > > We thank the reviewer for the clarification. We appreciate the guidance to focus on a literature discussion regarding the theoretical connections between Hard Negative Mining and temperature modulation, rather than providing an empirical comparison. We provide a detailed comparison below.
> > > >
> > > > **Hard Negative Mining vs. Temperature Modulation.** While temperature modulates the impact of negatives within a batch, Hard Negative Mining (HNM) explicitly intervenes at the data level to alter the sampling distribution. Early HNM approaches in contrastive learning focused on sampling strategies to bias the training data toward difficult examples (Robinson et al., 2020) [3], effectively decoupling the mining process from the loss function.
> > > >
> > > > The relationship between HNM and temperature is complementary rather than redundant. Wang & Liu (2021) [1] proved that low-temperature InfoNCE behaves asymptotically like a triplet loss with implicit hard negative mining. However, relying solely on a scalar $\tau$ couples the alignment of positive pairs with the repulsion of negative pairs. To resolve this, Yeh et al. (2021) [2] proposed decoupled learning by removing the positive term from the denominator, allowing for independent control over positive alignment and negative mining. Consequently, while HNM ensures the model is exposed to difficult samples, adaptive temperature mechanisms (Huang et al., 2023 [5]; Manna et al., 2025 [4]) are required to strictly calibrate their contribution to the gradient, preventing gradient explosion on outliers while maintaining focus on the decision boundary.
> > > >
> > > > We will include this discussion in the main text.
> > > >
> > > > [1] F. Wang and H. Liu, “Understanding the Behaviour of Contrastive Loss,” Mar. 20, 2021, arXiv: arXiv:2012.09740. doi: 10.48550/arXiv.2012.09740.
> > > >
> > > > [2] C.-H. Yeh, C.-Y. Hong, Y.-C. Hsu, T.-L. Liu, Y. Chen, and Y. LeCun, “Decoupled Contrastive Learning,” July 30, 2022, arXiv: arXiv:2110.06848. doi: 10.48550/arXiv.2110.06848.
> > > >
> > > > [3] J. Robinson, C.-Y. Chuang, S. Sra, and S. Jegelka, “Contrastive Learning with Hard Negative Samples,” Jan. 24, 2021, arXiv: arXiv:2010.04592. doi: 10.48550/arXiv.2010.04592.
> > > >
> > > > [4] S. Manna, S. Chattopadhyay, R. Dey, S. Bhattacharya, and U. Pal, “Dynamically Scaled Temperature in Self-Supervised Contrastive Learning,” May 10, 2024, arXiv: arXiv:2308.01140. doi: 10.48550/arXiv.2308.01140.
> > > >
> > > > [5] Z. Huang et al., “Model-Aware Contrastive Learning: Towards Escaping the Dilemmas,” June 11, 2023, arXiv: arXiv:2207.07874. doi: 10.48550/arXiv.2207.07874.

---

### Official Review · Reviewer_f5Mr · 2025-11-01

**Soundness:** 3
**Presentation:** 3
**Contribution:** 2
**Rating:** 4
**Confidence:** 4

**Summary:**

The paper proposes TeMo (Temperature Modulation), a new framework for multimodal contrastive learning that dynamically adjusts the temperature parameter on a per-pair basis according to sample similarity. TeMo leverages a similarity-driven temperature field to modulate both multimodal (image–text) and unimodal (image–image, text–text) contrastive losses, enabling more fine-grained representation alignment. Experimental results on various public benchmark datasets demonstrate consistent improvements over the InfoNCE, TS, and SLIP baselines.

**Strengths:**

- Adaptive temperature modulation: The paper proposes a per-pair, similarity-based temperature adjustment that jointly handles multimodal and unimodal contrastive learning, offering finer control of representation learning and reducing the modality gap.

- Empirical clarity and ablation support: The paper provides thorough ablation studies that isolate the effect of each component—multimodal modulation, unimodal losses, and scheduling—clearly demonstrating their individual and combined contributions

**Weaknesses:**

- Limited horizontal comparison with other contrastive training SOTA methods: The paper only contrasts TeMo against InfoNCE and Temperature Schedules (TS), without benchmarking against stronger contrastive learning variants such as SoftCLIP (Gao et al., AAAI 2024) or CWCL (Cross-Weighted Contrastive Learning, 2024). Since CLIP can be viewed as an InfoNCE-based baseline, the relative gain from temperature modulation seems smaller than the other type of contrastive loss SOTA approaches. See the performance reported in SoftCLIP Table 5 (SoftCLIP vs CLIP) or CWCL Table 3 (CWCL vs LiT on MSCOCO). The authors should include or at least discuss such horizontal comparisons to justify why temperature modulation remains a competitive or complementary direction compared to other fine-grained contrastive learning approaches.

- Potential manual tuning of τ parameters: Equation (4) λ(x̄, ȳ) = τ_min + τ_α * √sim(x, y) implies the proposed framework requiring manual tuning rather than being learned jointly with the model, this introduces additional hyperparameter sensitivity and weakens the claim of “adaptive” modulation.

- Narrow data scale and uncertain generalization: The study pretrains TeMo only on small- to mid-scale datasets (CC3M and CC12M), making it unclear whether the temperature modulation remains effective on larger or more diverse multimodal corpora (e.g., LAION, YFCC). The method may also require dataset-specific hyperparameter tuning (e.g., τ_min, τ_α), which could limit its practicality in broader pretraining settings.

**Questions:**

1. Can the authors compare TeMo with other fine-grained contrastive losses (e.g., SoftCLIP, CWCL) under the same experimental setup to clarify the relative contribution of temperature modulation?
2. Could temperature modulation be combined with SoftCLIP or CWCL, and would such integration yield additive or complementary gains?

---

> ### Author Response · Authors · 2025-11-23
>
> We thank the reviewer for the time taken to provide detailed and constructive feedback to our work. We appreciate that the reviewer acknowledges that our method offers “finer control of representation learning and reducing the modality gap” and appreciates the paper’s  “empirical clarity and ablation support.”
>
> In the following, we would like to address the reviewer's remaining concerns.
>
> > **W1, Q1. Limited horizontal comparison with other contrastive training SOTA methods: The paper only contrasts TeMo against InfoNCE and Temperature Schedules (TS), without benchmarking against stronger contrastive learning variants such as SoftCLIP (Gao et al., AAAI 2024) or CWCL (Cross-Weighted Contrastive Learning, 2024). Since CLIP can be viewed as an InfoNCE-based baseline, the relative gain from temperature modulation seems smaller than the other type of contrastive loss SOTA approaches. See the performance reported in SoftCLIP Table 5 (SoftCLIP vs CLIP) or CWCL Table 3 (CWCL vs LiT on MSCOCO). The authors should include or at least discuss such horizontal comparisons to justify why temperature modulation remains a competitive or complementary direction compared to other fine-grained contrastive learning approaches.**
>
> We thank the reviewer for suggesting these horizontal comparisons. We agree that benchmarking against state-of-the-art contrastive learning variants like SoftCLIP (Gao et al., AAAI 2024) and CWCL (2024) is essential to fully contextualize our contributions.
>
> We compare our approach with SoftCLIP on the ImageNet-1k zero-shot benchmark. For fairness, we use the official results reported in their paper, which employ the same pretraining dataset (CC3M) and architecture (ViT-B/16). It is important to note that differences in experimental environments and hyperparameters lead to slight variations in the CLIP baseline between our work and SoftCLIP. To account for this discrepancy, we focus our analysis on absolute performance gains. SoftCLIP achieves a 2.00 percentage points gain over its CLIP baseline, while our method attains a 2.23 percentage points gain. This suggests that our TeMo framework is more effective in enhancing the semantic structure of embeddings compared to SoftCLIP.
>
> For CWCL, since the official implementation is not publicly available, we re-implemented the method within our own codebase to ensure a strictly fair comparison under identical training conditions used for TeMo. Following the CWCL methodology, we froze the vision encoder and trained the model to align the text encoder with it. As shown in the experimental results below, TeMo consistently outperforms CWCL across all evaluation metrics.
>
> | Method | $\tau$ | COCO TR - R@1 |  COCO TR - R@5 |  COCO TR - R@10 | COCO IR - R@1 | COCO IR - R@5 | COCO IR - R@10 | R-Mean | Flickr TR - R@1 | Flickr TR - R@5 | Flickr TR - R@10 | Flickr IR - R@1 | Flickr IR - R@5 | Flickr IR - R@10 | R-Mean |
> | :--- | :---: | :---: | :---: | :---: | :---: | :---: | :---: | :---: | :---: | :---: | :---: | :---: | :---: | :---: | :---: |
> | InfoNCE | 0.01 | 21.64 | 45.77 | 57.47 | 28.60 | 53.90 | 66.56 | 45.66 | 42.12 | 68.84 | 78.76 | 53.60 | 81.30 | 88.90 | 68.92 |
> | CWCL | 0.01 | 21.40 | 46.32 | 58.82 | 14.26 | 34.40 | 45.99 | 36.87 | 37.90 | 67.40 | 78.20 | 27.64 | 55.54 | 67.84 | 55.36 |
> | CWCL | 0.05 | 22.18 | 47.04 | 59.16 | 14.51 | 34.57 | 45.96 | 37.24 | 40.40 | 67.40 | 80.60 | 28.60 | 56.56 | 67.76 | 56.89 |
> | **TeMo** | 0.01 - 0.05 | **23.28** | **47.48** | **59.62** | **30.36** | **56.82** | **67.92** | **47.58** | **44.16** | **73.26** | **81.84** | **56.00** | **82.70** | **90.80** | **71.46** |
>
> We will include extended comparison with both methods.
>
> > **W2 and part of W3. Potential manual tuning of τ parameters: Equation (4) λ(x̄, ȳ) = τ_min + τ_α * √sim(x, y) implies the proposed framework requiring manual tuning rather than being learned jointly with the model; this introduces additional hyperparameter sensitivity and weakens the claim of “adaptive” modulation.**
>
> We would like to clarify the definition of "adaptive" in our context and address the concern regarding hyperparameter sensitivity. The term "adaptive" in our work refers to dynamic, instance-level modulation, where the temperature is computed online based on the semantic similarity of the current pair (Equation 4), rather than being a static global hyperparameter. It does not imply that the boundary parameters ($\tau_{min}$​, $\tau_{\alpha}$​) are learned via gradient descent.
> Regarding the manual tuning of Equation (4), we emphasize that the process is straightforward and robust: we do not tune $\tau_{min}$. Instead, we adopt standard default values used in established literature and codebases (typically 0.01 or 0.07). In all reported experiments, we simply set $\tau_{min}​=0.01$, following the default configuration of the baseline repository. Consequently, the only parameter subject to selection is $\tau_{\alpha}$.​

---

> > ### Author Response · Authors · 2025-11-23
> >
> > > **W3. Narrow data scale and uncertain generalization: The study pretrains TeMo only on small- to mid-scale datasets (CC3M and CC12M), making it unclear whether the temperature modulation remains effective on larger or more diverse multimodal corpora (e.g., LAION, YFCC). The method may also require dataset-specific hyperparameter tuning (e.g., $\tau_{min}$, $\tau_{\alpha}$), which could limit its practicality in broader pretraining settings.**
> >
> > We agree that evaluating on massive-scale datasets, such as LAION-400M or YFCC, would provide a broader assessment of the method's scalability. However, such experiments require computational resources that exceed our resource capacity. Therefore, our results with TeMo should be viewed as competitive under constrained settings, rather than directly comparable to large-batch, large-data methods.
> >
> > > **Q2. Could temperature modulation be combined with SoftCLIP or CWCL, and would such integration yield additive or complementary gains?**
> >
> > We appreciate the reviewer’s interesting suggestion regarding the potential application of  our method to approaches like SoftCLIP or CWCL.
> >
> > While integrating these methods is theoretically feasible, we anticipate that it would introduce significant challenges regarding optimization and convergence. Since our proposed temperature modulation already actively regulates the gradient dynamics, effectively controlling the magnitude of the "pushing" and "pulling" forces between samples, combining it with other mechanisms that also modify the loss landscape could lead to interference.
> >
> > Specifically, combining multiple dynamic weighting schemes may result in conflicting gradient signals, making hyperparameter tuning notably difficult. Therefore, while such integration could theoretically yield complementary gains, we believe the complexity of balancing these interacting components is non-trivial and best reserved for future investigation.

---

> ### Comment · Reviewer_f5Mr · 2025-11-27
>
> Thank you for the authors’ additional experiments and clarifications. My remaining concern relates to the scale of the training data, which appears to be limited to approximately 12M image–text pairs. In prior large-scale contrastive learning works such as LiT (Zero-Shot Transfer with Locked-image Text Tuning), Figure 1 clearly shows that contrastive learning benefits significantly from purely data-driven training over substantially larger corpora, typically on the order of 250M image–text pairs, to achieve robust semantic alignment and meaningful representation learning. Similarly, CWCL (Cross-Modal Transfer with Continuously Weighted Contrastive Loss) also reports training on datasets exceeding 150M image–text pairs, as shown in their Figure 1.
>
> While the proposed formulation $\lambda(\bar{x}, \bar{y}) = \tau_{\min} + \tau_{\alpha}\sqrt{\mathrm{sim}(x, y)}$
> introduces a more nuanced adjustment of similarity by modulating the effective temperature via $\tau_{\alpha}$ manually with proper selection, and may indeed yield improved learning efficiency at smaller scales, it remains unclear whether this advantage persists at large-scale regimes. In particular, it would be important to understand whether TeMo continues to outperform or remain competitive when scaled to >150M or even 250M image–text pairs, where representation learning dynamics and negative sample diversity differ substantially. Specifically, after prior art's contrastive learning methods have already learned strong representations from massive data distributions, it remains uncertain whether TeMo’s temperature modulation continues to provide meaningful benefits once the model has already achieved high-quality semantic representations through large-scale data-driven contrastive learning.
> ⁡
> I therefore maintain my original score for now. I will also revisit the discussion later to consider any further value points raised by other reviewers.

---

> > ### Author Response · Authors · 2025-12-03
> >
> > We thank the reviewer for their continued engagement and for acknowledging our additional experiments. We appreciate the rigorous perspective regarding the scalability of contrastive learning methods. To ensure a precise comparison with prior works, we wish to clarify the specific metrics used in the cited works regarding training scale.
> >
> > 1. **Clarification on "Scale" (Throughput vs. Unique Data):** We note that the scalability figures in both CWCL and LiT (e.g., CWCL Fig. 1, LiT Fig. 1) plot performance against "Image-text pairs seen" (total throughput over all epochs), rather than the count of unique images in the dataset.
> >     * **CWCL:** This method trains on CC12M and YFCC15M. While the x-axis extends to "200M+," this reflects total samples processed, not a unique dataset of 200M images. The actual unique data size is approximately ~25M pairs.
> >     * **LiT:** Similarly, Figure 1 tracks "Total pairs seen." Even when benchmarking on the YFCC100M-subset (~15M unique images), the x-axis extends far beyond this number because it accounts for cumulative epochs.
> > 2. **TeMo’s Effective Scale (360M Samples):** When viewed through this metric of total throughput, our experiments are highly competitive. Training on CC12M (12M unique pairs) for 30 epochs yields a cumulative total of 360M samples seen. This volume significantly exceeds the "150M" or "200M" sample benchmarks. Thus, TeMo demonstrates robustness and convergence at a computational regime comparable to the effective duration of these state-of-the-art methods.
> >
> > 3. **Extended Validation Plan:** To further validate scalability on larger unique data distributions, we commit to including experiments on the combined YFCC15M + CC12M dataset (~25M unique pairs) in the final version of the paper. This aligns directly with the unique data scale of CWCL. While the rebuttal timeline prevents the completion of this large-scale run, the consistent performance gains observed when scaling from 3M to 12M provide strong evidence that these benefits will generalize to the larger split.
> >
> > We will include this discussion in the main text to better contextualize our experiments.

---

### Official Review · Reviewer_81f6 · 2025-11-03

**Soundness:** 3
**Presentation:** 3
**Contribution:** 2
**Rating:** 4
**Confidence:** 4

**Summary:**

This paper introduces TeMo (Temperature Modulation), a novel framework for multimodal contrastive learning (CL) that adaptively adjusts the temperature parameter $\tau$ on a per-pair basis. Traditional contrastive learning methods such as CLIP, SimCLR, and MoCo typically use a global or learnable scalar temperature shared across all sample pairs. This uniform temperature limits the model’s ability to balance repulsive forces among diverse semantic classes—especially in long-tailed or multimodal datasets.

To address this limitation, the authors propose a similarity-based temperature modulation mechanism, where the temperature $\tau_{ij}$ for each positive–negative pair is dynamically modulated according to their similarity score. This allows finer control of the InfoNCE loss. In addition to adaptive temperatures for the multimodal objective, TeMo incorporates temperature-modulated unimodal contrastive losses that help refine the local structure within each modality (e.g., image–image or text–text). The framework uses a progressive training schedule, starting with a lower temperature (capturing fine-grained, instance-level semantics) and gradually increasing it to encourage coarse, semantic-level alignment.

**Strengths:**

**1. Novel temperature modulation framework:** The paper introduces TeMo, a principled framework that adaptively modulates the temperature parameter on a per-pair basis according to similarity scores. This is a meaningful advance over prior works that rely on a fixed or globally learned temperature. The method provides finer control over the contrastive learning process, improving representation quality across both unimodal and multimodal settings. This idea is conceptually simple yet broadly applicable to various contrastive learning architectures (e.g., CLIP-style models).

**2. Comprehensive empirical evaluation and consistent performance gains:** The authors conduct extensive experiments on zero-shot retrieval (MSCOCO, Flickr30k) and zero-shot classification (CIFAR10, CIFAR100, ImageNet-1k), demonstrating consistent improvements over multiple temperature-adaptation baselines. The fact that TeMo outperforms across diverse datasets and training scales (CC3M and CC12M) indicates good generalization and practical utility of the approach.

**Weaknesses:**

Here are some concerns for this paper:

**1. Missing Discussion of Related Literature:** Some closely related literauter have not discussed in the paper. The paper overlooks several closely related studies that address temperature adaptation and modality imbalance in contrastive multimodal learning. For example, Manna et al. (2023) [1] propose a similarity-dependent temperature scaling function in the contrastive loss. Their method dynamically adjusts temperature based on the cosine similarity between samples, rather than using a fixed global $\tau$. Yaras et al. (2024) [2] proposed a conceptually similar approach that dynamically controls the temperature to mitigate the modality gap—the discrepancy between image and text embedding distributions. This work provides both theoretical and empirical insights into how temperature modulation affects multimodal alignment and balance.

[1] Manna, Siladittya, et al. "Dynamically Scaled Temperature in Self-Supervised Contrastive Learning." IEEE Transactions on Artificial Intelligence (2025).
[2] Yaras, Can, et al. "Explaining and Mitigating the Modality Gap in Contrastive Multimodal Learning." Conference on Parsimony and Learning, 2024.

**Lack of Principled Understanding:** While the proposed TeMo framework demonstrates promising empirical performance, the paper lacks a principled theoretical understanding of how and why temperature modulation improves multimodal contrastive learning. The method is primarily motivated from an intuitive perspective—that per-pair temperature scaling provides finer control over learning signals—but the paper does not formally analyze its effect on the InfoNCE loss landscape, gradient dynamics, or representation geometry.

**Questions:**

**1.** Can TeMo be connected to existing theoretical frameworks on temperature scaling and representation uniformity (e.g., Wang & Isola, ICML 2020)?

Wang, T. and Isola, P. (2020). "Understanding Contrastive Representation Learning through Alignment and Uniformity on the Hypersphere." International Conference on Machine Learning (ICML 2020).

**2.** How does modulating $\tau_{ij}$ based on pairwise similarity alter the effective margin or smoothness of the contrastive objective?

**3.** What guarantees, if any, can be made regarding stability or convergence when temperatures vary across pairs?

**Details Of Ethics Concerns:**

N.A.

---

> ### Author Response · Authors · 2025-11-23
>
> We thank the reviewer for the time taken to provide detailed and constructive feedback to our work. We appreciate that the reviewer acknowledges our novelty and broad applicability of our approach. Further, we are encouraged to find our empirical evaluation is comprehensive and performance gains are consistent.
>
> In the following, we would like to address the reviewer's remaining concerns.
>
> > **W1. Missing Discussion of Related Literature**
>
> We thank the reviewer for highlighting these relevant studies. Below we provide the extended discussion.
>
> 1. Empirical Comparison with Manna et al. (2023) (DySTreSS) [1]: To provide a rigorous comparison, we implemented the DySTreSS framework, adopting it to the multimodal contrastive setting. To ensure a fair and comprehensive evaluation, we tested multiple configurations:
>     * We evaluated both the "Shifted" and "Non-Shifted" versions of the DySTreSS loss.
>     * We experimented with the temperature range provided by DySTreSS [0.1, 0.2]  and the range used in our experiments [0.01, 0.05].
>     * Results: As shown in the table below, DySTreSS underperforms compared to the standard InfoNCE baseline and significantly lags behind our proposed TeMo method. The suboptimal performance of DySTreSS in this domain is consistent with our own ablation findings presented in the paper (Table 3, Row b)). We established that applying similarity-based temperature modulation in isolation—without a progressive scheduler to delay modulation and unimodal losses to enforce uniformity—fails to improve upon the baseline and can even degrade performance. DySTreSS lacks these stabilizing components. In contrast, TeMo succeeds because the modulation is integrated into a framework that first establishes a uniform embedding space (via scheduling) before refining alignment.
>
> | Method | $\tau$ | COCO TR - R@1 | COCO TR - R@5 | COCO TR - R@10 | COCO IR - R@1 | COCO IR - R@5 | COCO IR - R@10 | R-Mean | Flickr TR - R@1 | Flickr TR - R@5 | Flickr TR - R@10 | Flickr IR - R@1 | Flickr IR - R@5 | Flickr IR - R@10 | R-Mean |
> | :--- | :---: | :---: | :---: | :---: | :---: | :---: | :---: | :---: | :---: | :---: | :---: | :---: | :---: | :---: | :---: |
> | InfoNCE | 0.01 | 21.64 | 45.77 | 57.47 | 28.60 | 53.90 | 66.56 | 45.66 | 42.12 | 68.84 | 78.76 | 53.60 | 81.30 | 88.90 | 68.92 |
> | DySTreSS | 0.1 - 0.2 | 9.70 | 26.66 | 37.48 | 14.16 | 33.48 | 45.48 | 27.83 | 21.94 | 50.10 | 63.32 | 31.30 | 58.70 | 72.20 | 49.59 |
> | DySTreSS | 0.1 - 0.2 (Shifted) | 7.87 | 22.57 | 32.89 | 12.52 | 31.12 | 41.82 | 24.80 | 19.38 | 45.84 | 59.26 | 28.30 | 54.60 | 67.60 | 45.83 |
> | DySTreSS | 0.01 - 0.05 | 19.12 | 41.92 | 52.64 | 25.90 | 51.76 | 64.48 | 42.80 | 37.12 | 65.42 | 75.60 | 47.70 | 79.20 | 86.60 | 65.28 |
> | DySTreSS | 0.01 - 0.05 (Shifted) | 19.22 | 41.45 | 53.58 | 23.44 | 48.62 | 59.76 | 41.01 | 35.94 | 64.72 | 75.08 | 47.20 | 75.90 | 84.40 | 63.87 |
> | **TeMo** | 0.01 - 0.05 | **23.28** | **47.48** | **59.62** | **30.36** | **56.82** | **67.92** | **47.58** | **44.16** | **73.26** | **81.84** | **56.00** | **82.70** | **90.80** | **71.46** |
>
> 2. Discussion on Yaras et al. (2024) [2]: We agree that Yaras et al. provide valuable theoretical insights into how temperature modulation affects the modality gap. However, our approach distinguishes itself by explicitly tackling this gap through additional constraints rather than temperature alone.
>     * TeMo's Approach: We incorporate unimodal contrastive losses, which explicitly enforce uniformity within each modality. As shown in Figure 3, this regularization reduces the modality gap, providing a concrete solution that complements the theoretical analysis offered by Yaras et al.
>
> We will include this discussion in the main to better contextualize our contributions.
>
> [1] Manna, Siladittya, et al. "Dynamically Scaled Temperature in Self-Supervised Contrastive Learning." IEEE Transactions on Artificial Intelligence (2025).
>
> [2] Yaras, Can, et al. "Explaining and Mitigating the Modality Gap in Contrastive Multimodal Learning." Conference on Parsimony and Learning, 2024.

---

> ### Author Response · Authors · 2025-11-23
>
> > **W2 and Q1. Can TeMo be connected to existing theoretical frameworks on temperature scaling and representation uniformity (e.g., Wang & Isola, ICML 2020)?**
>
> We thank the reviewer for this insightful connection. We explicitly designed TeMo to leverage the theoretical trade-off between Alignment and Uniformity. Our method can be interpreted as a dynamic mechanism that finds a balance between the two properties.
>
> 1. Regulating the Trade-off via Adaptive $\tau$: As established by Wang & Liu (2021), a low temperature favors Uniformity, while a higher temperature facilitates Alignment. TeMo applies this principle at the instance level:
>     * For dissimilar pairs: We assign a lower $\tau$, enforcing sharper gradients that push samples apart to maximize Uniformity.
>     * For high-similarity pairs: We dynamically increase $\tau$. This relaxes the penalty for hard samples, prioritizing semantic Alignment and preventing the model from "over-separating" semantically related concepts.
> 2. Temporal Dynamics and Scheduling: Our progressive scheduler reflects a coarse-to-fine optimization strategy.
>     * Phase 1 (Uniformity): In the early stages, the scheduler maintains a fixed, low $\tau$. This forces the model to expand the embedding space and establish Global Uniformity, preventing representation collapse.
>     * Phase 2 (Alignment): As training progresses, the scheduler activates the modulation. This shifts the focus toward local Alignment, refining the semantic clusters established in Phase 1. Our ablation studies confirm this: enforcing strong structural alignment too early, before the embedding space is uniform, degrades performance (as seen in Table 3, Row b)), validating the need for this temporal evolution.
>
> We thank the reviewer again for highlighting this connection. We will incorporate the suggested references and the discussion above regarding the Alignment-Uniformity trade-off into the final manuscript.
>
> > **W2 and Q2. How does modulating $\tau_{ij}$ based on pairwise similarity alter the effective margin or smoothness of the contrastive objective?**
>
> We thank the reviewer for this insightful question regarding the geometric and optimization implications of our method.
>
> 1. Impact on Effective Margin: Regarding the effective margin, we refer the reviewer to Figure 4 in the main paper, which visualizes the density histograms of cosine similarities for positive and negative pairs. As shown in the figure, the baseline model exhibits a significant "overlap" region where the similarity distributions of hard negatives and positives intersect. This overlap indicates a small or non-existent margin, leading to classification ambiguity. In contrast, our proposed modulation explicitly pushes these distributions apart. By dynamically adjusting the temperature, the model applies a more nuanced penalty that effectively clears the "confusion zone," thereby increasing the effective margin between the positive and negative classes and resulting in a more separable representation. This enhanced separability directly improves robustness against false positives, as further demonstrated in Section G of the Supplementary Material.
> 2. Impact on Smoothness of the Objective: The proposed modulation $\lambda(x,y)$ acts as a data-dependent regularizer. In standard InfoNCE with a fixed low temperature, high-similarity negative pairs produce extremely sharp gradients, creating "cliffs" in the loss landscape that can destabilize training. Our method increases the temperature $\tau$ as similarity increases. This locally "softens" the softmax distribution for these high-similarity pairs, effectively dampening the gradient magnitude and preventing explosive updates from hard negatives.
>
> We thank the reviewer again for this insightful question. We will incorporate the above discussion—specifically regarding the effective margin and the smoothness of the contrastive objective—into the final manuscript.
>
> > **Q3. What guarantees, if any, can be made regarding stability or convergence when temperatures vary across pairs?**
>
> We address the concern regarding stability through structural design and empirical evidence:
> 1. Bounded Gradient Scaling: TeMo strictly bounds every per-pair temperature $\tau_{ij} \in [\tau_{min}, \tau_{min} + \tau_{\alpha}]$, the logit scaling is strictly upper-bounded, preventing gradient explosion.
> 2. Empirical Stability: As shown in Figure 2, batch-level temperatures converge smoothly and are insensitive to LR scheduling.
> 3. Context: We note that standard contrastive objectives (such as InfoNCE) used with deep networks also lack strict theoretical convergence guarantees due to the non-convex landscape, yet are widely adopted for their empirical stability. Our method inherits this stability while improving representation alignment.

---

### Author Response · Authors · 2025-12-03

**Summary of the contributions of the paper**

We summarize our contribution:
1. We propose TeMo, a novel temperature modulation framework for multimodal contrastive learning, which introduces a per pair temperature modulation based on the similarity of each pair for both unimodal and multimodal losses, enabling more precise control over the learned representation space.
2. We show that combining multimodal and unimodal contrastive losses is particularly effective when used with our temperature modulation approach.
3. We provide an in-depth evaluation of the characteristics of the proposed system and show that TeMo outperforms prior temperature adaptation approaches on 21 zero-shot retrieval and classification benchmarks.

**Summary of the initial reviews**

**Reviewer 81f6**

* **Strengths**: Highlighted the novelty of the per-pair temperature modulation and the comprehensive empirical evaluation over different tasks (zero-shot retrieval and classification) and training scale (CC3M and CC12M).

* **Weaknesses**: Noted the absence of comparisons with specific related works (Manna et al. [1] and Yaras et al. [2]) and the lack of a theoretical explanation for the method's efficacy.

* **Key Questions**: Requested clarification on the method's theoretical links to established "Alignment vs. Uniformity" frameworks (Wang et al. [3]). The reviewer further inquired about the method's influence on the effective margin and gradient smoothness, as well as its associated convergence guarantees.

**Reviewer f5Mr**

* **Strengths**: Highlighted the finer control over representation learning, the reduction of the modality gap using, and the detailed component analysis.

* **Weaknesses**: Pointed out the limited horizontal comparison with other contrastive training SOTA methods (Y. Gao et al. [1], R. S. Srinivasa et al. [2]), raised concerns regarding the training scale, and the manual tuning of the parameters on Equation (4).

* **Key Questions**: Requested empirical comparisons against SoftCLIP [4] and CWCL [5], and asked whether TeMo can be used as a complementary component to these methods.

**Reviewer sR36**

* **Strengths**: Identified the simplicity and ease of implementation as positive aspects.

* **Weaknesses**: Argued that performance gains were likely attributable to the "Unimodal Loss" rather than the proposed modulation. Additionally, stated that the zero-shot evaluations were insufficient and noted a lack of comparison to hard negative mining strategies.

* **Key Questions**: The reviewer didn’t provide any questions.

---

> ### Author Response · Authors · 2025-12-03
>
> **Summary of discussions:**
>
> **Rebuttal for Reviewer 81f6 (Initial rating 4, updated score -)**
> * **Actions**:
>     * We implemented the requested baseline (DySTreSS) and demonstrated that TeMo outperforms them under the same training configuration and contextualized Yaras et al., explaining how our unimodal losses explicitly target the modality gap they describe.
>     * We linked TeMo to "Alignment vs. Uniformity," clarifying how our scheduler first establishes global uniformity (low $\tau$) before refining local alignment (modulated $\tau$).
>     * We demonstrated that TeMo increases the effective margin (reducing overlap between positive/negative distributions) and ensures smoothness by dampening explosive gradients. We further addressed stability by confirming that our temperature scaling is strictly bounded.
>
> * **Outcome**: No response from the reviewer.
>
> **Rebuttal for Reviewer f5Mr (Initial rating 4, updated score - interrupted discussion)**
>
> * **Actions**:
>     * We conducted direct comparisons against the requested methods. For SoftCLIP [4], TeMo achieves a higher absolute performance gain  (+2.23% vs +2.00%).
>
>     * For CWCL [5], we re-implemented the method (as official code was unavailable) and demonstrated that TeMo consistently outperforms it across all retrieval metrics.
>
>     * We clarified that our modulation is "adaptive" at the instance level rather than a learned parameter, and requires minimal tuning.
>
> * **Outcome** : The reviewer acknowledged the extensive new experiments and clarifications. In the final comment, the remaining concern relates to the scale of the training data relative to CWCL (see Fig. 1 in CWCL) and LiT (see Fig. 1 in LiT). In the response, it is clarified that these methods report scale in terms of throughput data rather than unique data. Throughput data reflects the total number of samples processed during training and does not correspond directly to the size of the underlying dataset. Following this clarification, we provide an additional plot to demonstrate robustness with respect to throughput data, consistent with the reporting practices of CWCL and LiT.
> At the moment of the interrupted discussion, the reviewer retained their score and promised to revisit the discussion later.
>
> **Rebuttal for Reviewer sR36 (Initial score 2, updated score 6)**
> * **Actions:**
>
>     * We performed the requested ablation experiment: "Base Model + Unimodal Loss" without modulation. We observed that the results were below our InfoNCE baseline, showing the difficulty in balancing the cross modal and unimodal losses without temperature modulation.
>
>     * We expanded the evaluation to 17 new zero-shot benchmarks, with TeMo achieving the best performance.
>     * We expanded the discussion regarding potential connection between hard-negative and temperature modulation.
>
> * **Outcome:** The reviewer increased their score from 2 to 6, acknowledging the validity of the contributions.
>
> **Summary of improvements due to the discussions:**
> * Provided comparisons against other contrastive learning methods, such as DySTreSS, SoftCLIP and CWCL, showing that TeMo consistently outperforms them.
> * Expanded the zero-shot classification performance to include 17 additional datasets, demonstrating robust generalization.
> * Added specific ablation results isolating the impact of the unimodal losses, empirically proving that temperature modulation is the primary driver of performance.
> * Argued how TeMo regulates the effective margin, gradient smoothness to improve training stability and how it connects to  "Alignment vs. Uniformity."
>
> **Concluding remarks:**
>
> Given the clear empirical validation of our contributions (consistent improvements over state-of-the-art baselines) and the comprehensive resolution of technical questions (demonstrating robust generalization and clarifying theoretical grounding), we believe TeMo offers a valuable contribution to the community. We are grateful for the AC's time and consideration.
>
> Kind regards,
>
> Authors of Submission **9080**
>
> **References:**
>
> [1] Manna, Siladittya, et al. "Dynamically Scaled Temperature in Self-Supervised Contrastive Learning." IEEE Transactions on Artificial Intelligence (2025).
>
> [2] Yaras, Can, et al. "Explaining and Mitigating the Modality Gap in Contrastive Multimodal Learning." Conference on Parsimony and Learning, 2024.
>
> [3] Wang, T. and Isola, P. (2020). "Understanding Contrastive Representation Learning through Alignment and Uniformity on the Hypersphere." International Conference on Machine Learning (ICML 2020).
>
> [4] Y. Gao et al., “SoftCLIP: Softer Cross-modal Alignment Makes CLIP Stronger,” Dec. 16, 2023, arXiv: arXiv:2303.17561. doi: 10.48550/arXiv.2303.17561.
>
> [5] R. S. Srinivasa et al., “CWCL: Cross-Modal Transfer with Continuously Weighted Contrastive Loss,” Sept. 26, 2023, arXiv: arXiv:2309.14580. doi: 10.48550/arXiv.2309.14580.

---

### Meta-Review · Area_Chair_bsH1 · 2025-12-22

**Summary:**

This paper proposes TeMo, a novel temperature modulation framework for multimodal contrastive learning. The key idea is to apply pair-specific temperature modulation based on the similarity of each sample pair, consistently across both unimodal and multimodal contrastive objectives. This design enables finer-grained control of the representation geometry during training.

The authors further demonstrate that jointly optimizing unimodal and multimodal contrastive losses is particularly effective when coupled with the proposed temperature modulation strategy. Extensive empirical analysis is provided to characterize the behavior of the method. Across 21 zero-shot retrieval and classification benchmarks, TeMo consistently outperforms existing temperature adaptation approaches, indicating strong robustness and generalization.

**Reviewer Concerns:**

Reviewer Concerns are still outstanding:

(1) Lack of Principled Understanding: While the proposed TeMo framework demonstrates promising empirical performance, the paper lacks a principled theoretical understanding of how and why temperature modulation improves multimodal contrastive learning.

(2) Potential manual tuning of $\tau$ parameters (Although the authors note that the method is effective when the temperature parameter is set to a fixed constant, the robustness and reliability of this choice require stronger justification. In particular, it remains unclear under what conditions a single constant value is sufficient, or when such a choice may fail. Additional empirical evidence or analysis examining the sensitivity to would further strengthen the claims.)

**Reviewer Scores:**

Reviewer 81f6 (score: 4) is unlikely to change their score due to the unresolved Concern (1).

Reviewer f5Mr (score: 4) is similarly unlikely to change their score due to the unresolved Concern (2).

Reviewer sR36 (initial score: 2) has indicated an intention to increase the score to 6.

---

### Decision · Program_Chairs · 2026-01-26

Reject